# ProMoS: Prototype-Guided Distillation for Generalist Graph Anomaly Detection

## Abstract

Graph anomaly detection (GAD) is crucial in high-stakes domains. Recently, generalist GAD is a type of GAD that trains a single detector and can be transferred to new graphs, and has attracted attention. However, existing methods often rely on scarce and costly annotations for training and sometimes even require few-shot support at inference, which limits their robustness to diverse and unseen anomaly patterns. To address this limitation, we introduce ProMoS, the first unsupervised generalist GAD framework, which detects anomalies by modeling the abundant normality in unlabeled data. Specifically, we introduce a knowledge-distillation (KD) architecture that distills normality representations from a frozen self-supervised graph neural network (GNN) teacher to a mixture-of-students (MoS) model. The MoS employs a shared branch to capture global patterns and a lightweight personalized branch to extract local normality from the teacher, avoiding learning normality from scratch while improving both expressiveness and efficiency. Second, we propose prototype-guided soft-label distillation to align the student with the teacher in a shared prototype space, thereby improving cross-graph transferability and generalizability. During inference, ProMoS performs zero-shot anomaly detection on unseen graphs based on teacher-student distillation bias and prototype geometric deviation. Extensive experiments on eleven zero-shot GAD tasks show that ProMoS consistently outperforms state-of-the-art supervised, unsupervised, and generalist baselines while reducing computational overhead, charting a practical path toward label-free, zero-shot generalist GAD. [1]

## 1 Introduction

Graph anomaly detection (GAD) has attracted attention in high-stakes domains modeled with graphs (Ma et al., 2021), such as finance (Yuan et al., 2025), social networks (Xu et al., 2022a), and cybersecurity (Wang & Zhu, 2022), by identifying deviations from normal patterns automatically (Wang et al., 2025; Li et al., 2025). Despite recent progress, conventional GAD methods (Liu et al., 2021; Tang et al., 2022; Qiao & Pang, 2023) require retraining and extensive hyperparameter tuning to handle each new coming graph, incurring prohibitive computational and operational costs that are untenable for large-scale or latency-sensitive applications (Niu et al., 2025). To overcome these limitations, recent studies explore generalist GAD models that are trained once and generalize across graphs without any retraining on the target graph (Liu et al., 2024).

Although effective in some scenarios, existing generalist GAD methods still heavily rely on labeled supervision (Liu et al., 2024; Niu et al., 2025; Qiao et al., 2025a). In practice, anomaly labels are scarce, costly (Ma et al., 2024), and intrinsically unable to cover the constantly evolving space of abnormal behavior in the open world, since no fixed annotation set can fully enumerate this variability (Sricharan & Das, 2014; Wang et al., 2019). In contrast, large graphs naturally provide abundant unlabeled signals (Liu et al., 2022; Xie et al., 2022): normal nodes dominate the graph and collectively harbor rich and robust normal patterns of behavior and connectivity that reflect the underlying regularities of the graph (Cao et al., 2025), with anomalies emerging as significant deviations from these patterns. Motivated by these insights, we ask whether a *generalist* GAD model can be trained once in an *unsupervised* manner and then transferred across graphs.

---

[1]The source code and datasets are available at: https://anonymous.4open.science/r/ProMoS

Table 1: A brief comparison of representative GAD methods. "Zero-shot" refers to inference on unseen graphs without using any labeled data for fine-tuning or adaptation; SSL refers to graph self-supervised learning (pre-trained) models.

| Method | Unsupervised | Zero-shot | Generalist | Training Paradigm |
|---|:---:|:---:|:---:|---|
| DOMINANT (Ding et al., 2019) | ✓ | ✓ | ✗ | Train from scratch |
| CoLA (Liu et al., 2021) | ✓ | ✓ | ✗ | Train from scratch |
| TAM (Qiao & Pang, 2023) | ✓ | ✓ | ✗ | Train from scratch |
| ARC (Liu et al., 2024) | ✗ | ✗ | ✓ | Train from scratch |
| UNPrompt (Niu et al., 2025) | ✗ | ✓ | ✓ | Train from scratch |
| AnomalyGFM (Qiao et al., 2025a) | ✗ | ✓ | ✓ | Train from scratch |
| **ProMoS (Ours)** | ✓ | ✓ | ✓ | **Transferred from SSL** |

Realizing this vision is non-trivial and poses two key challenges. ❶ *Comprehensive normality modeling*. A fundamental challenge is to learn comprehensive and representative normal patterns without labeled data (Ma et al., 2021; Qiao et al., 2025b). An ill-designed unsupervised objective may recover a narrow and unrepresentative manifold of normal patterns from the data. Leveraging such a biased characterization of normality for anomaly detection inevitably leads to degraded performance (Cao et al., 2025). ❷ *Cross-graph heterogeneity gap*. Significant discrepancies exist across graphs in both node-attribute semantics and topological characteristics, posing a critical challenge to cross-graph transfer. For instance, financial transaction graphs contain scalar fields like monetary amounts and exhibit hub-centric structures, whereas social networks contain unstructured text such as posts and bios, while exhibiting community-centric connectivity (Ruff et al., 2021). The prevailing unsupervised GAD methods primarily rely on objectives like within-graph instance discrimination (Liu et al., 2021; Xu et al., 2025) or feature reconstruction (Ding et al., 2019; Zou et al., 2024), tend to overfit to dataset-specific fine-grained patterns. This overfitting severely hinders the generalization of learned normal patterns to graphs with different data distributions.

To tackle the above challenges, we propose **ProMoS**, a **Pro**totype-guided **M**ixture-**o**f-**S**tudents framework for unsupervised generalist GAD, as shown in Table 1. Specifically, to address Challenge ❶, we propose a knowledge distillation (KD) framework that distills normality priors from a well-trained self-supervised GNN into the student module, departing from the conventional practice of learning normal patterns from scratch. To balance expressiveness and efficiency, the student module devises a mixture-of-students (MoS) architecture, with a shared branch capturing global regularities and a sparsely activated personalized branch modeling diverse local normal patterns. To address Challenge ❷, we propose prototype-guided soft-label distillation, which aligns teacher and student predictions with a set of learnable semantic prototypes initialized by clustering features, thereby avoiding reliance on instance-level or feature-level fine-grained modeling. To further enhance transferability, we introduce a discrepancy-aware commitment and refinement objective that uses sample reliability-weighted to enforce stability in the teacher's semantic space across graphs while continually updating prototypes to encode higher-quality transferable semantics. Theoretically, we prove that the expected prediction error of the MoS framework is no greater than that of any individual student, ensuring its effectiveness. During inference, ProMoS requires *no* retraining or fine-tuning. Anomaly scores are derived by combining two complementary signals: prototype-level distillation bias and geometric deviation, enabling robust zero-shot detection on unseen graphs. Our contributions are summarized as follows:

• We propose the *first* unsupervised generalist GAD framework, ProMoS, which eliminates the reliance on labeled data for cross-graph generalization. Our work establishes a new pathway to take full advantage of large-scale unlabeled graph data for efficient and scalable anomaly detection.

• We propose a novel unsupervised KD framework for generalist GAD that transfers priors from a pre-trained graph SSL teacher and introduces a MoS module to balance expressiveness and efficiency. A tailored loss suite, comprising prototype distillation and discrepancy-aware commitment and refinement, further improves cross-graph generalizability.

• Extensive experiments on 11 real-world graphs demonstrate that ProMoS achieves superior generalization over state-of-the-art supervised, unsupervised, and generalist GAD baselines, while incurring lower computational overhead.

## 2 RELATED WORK

**Anomaly Detection on Graphs.** The rapid progress of GNNs has substantially advanced research on GAD (Ma et al., 2021). Existing GNN-based GAD approaches can be broadly categorized into supervised and unsupervised methods. Supervised methods (Tang et al., 2022; Gao et al., 2023; Dong et al., 2025) rely on limited anomaly labels to learn explicit decision boundaries, but often suffer from overfitting to seen anomalies and tend to misclassify unseen anomalies as normal (Wang et al., 2025). In contrast, unsupervised methods have gained increasing attention for their label-agnostic nature (Qiao et al., 2025b), typically leveraging reconstruction errors (Ding et al., 2019; Roy et al., 2024; Zou et al., 2024) or contrastive (Liu et al., 2021; Wang et al., 2023) proxy tasks to model normal patterns and detect anomalies. While both paradigms achieve promising results under the conventional setting where training and inference are performed on the same graph, they typically break down under cross-graph settings, revealing a critical generalization gap (Liu et al., 2024).

**Generalist Anomaly Detection.** Generalist anomaly detection has recently gained traction as a promising direction for addressing the challenges of label scarcity and poor model generalization in anomaly detection (Yao et al., 2024). Inspired by advances in image anomaly detection (Zhu & Pang, 2024), several early studies have explored supervised generalist GAD models and demonstrated initial effectiveness (Liu et al., 2024; Niu et al., 2025; Qiao et al., 2025a). However, these approaches typically rely on extensive labeled data to learn transferable representations and domain knowledge. For example, ARC (Liu et al., 2024) requires substantial annotations to train not only the encoder but also its context module, and still depends on a few target-domain samples during inference. UNPrompt (Niu et al., 2025) utilizes label-driven soft prompts, while AnomalyGFM (Qiao et al., 2025a) explicitly constructs class-specific priors for normal and anomalous categories. In contrast to these label-intensive methods, we take the first step toward unsupervised generalist GAD, aiming to learn cross-graph transferable normality patterns without any annotations. Our framework provides a new perspective for achieving zero-shot anomaly detection across diverse graphs.

**Knowledge Distillation on Graphs.** Knowledge distillation (KD) (Hinton et al., 2014) was first introduced to facilitate model compression and the creation of resource-efficient architectures. Later, it was extended to various domains, including image and video anomaly detection (Georgescu et al., 2021; Zhang et al., 2023). Graph KD has gained traction in recent years (Tian et al., 2025), yet its adoption in GAD remains limited (Qiao et al., 2025b). Most existing efforts target graph-level GAD tasks (Ma et al., 2022; Lin et al., 2023; Cai et al., 2024), while node-level GAD has received comparatively little attention. Moreover, prevailing methods adopt a *one-teacher–one-student* paradigm that aligns hidden states or logits, neglecting the design of student architectures and failing to address cross-graph generalization. In this work, we advance KD for GAD by (i) establishing its feasibility at the node-level GAD and (ii) introducing a mixture-of-students that aligns student outputs to teacher-derived prototype soft labels, improving cross-graph transferability and generalization.

## 3 METHODOLOGY

### 3.1 PRELIMINARIES

**Notation.** An attributed graph is denoted as $\mathcal{G} = (\mathcal{V}, \mathcal{E}, \mathbf{X})$ with a node set $\mathcal{V} = \{v_i\}_{i=1}^n$, en edge set $\mathcal{E} \subseteq \mathcal{V} \times \mathcal{V}$, and the node feature matrix $\mathbf{X} \in \mathbb{R}^{n \times d}$. Its topological structure is recorded in the adjacency matrix $\mathbf{A} \in \{0,1\}^{n \times n}$ where $A_{ij} = 1$ iff $(v_i, v_j) \in \mathcal{E}$. In GAD, binary labels $y_i \in Y \subset \{0,1\}^n$ split the nodes into a normal set $\mathcal{V}_n = \{v_i \,|\, y_i = 0\}$ and an anomalous set $\mathcal{V}_a = \{v_i \,|\, y_i = 1\}$ where $\mathcal{V}_n \cap \mathcal{V}_a = \varnothing$, $\mathcal{V}_n \cup \mathcal{V}_a = \mathcal{V}$, and $|\mathcal{V}_n| \gg |\mathcal{V}_a|$.

**Conventional GAD Setting.** Most existing GAD studies follow the *one-graph-one-model* protocol: a detector is trained and deployed on the same graph $\mathcal{G}$. Learning proceeds in either a supervised or unsupervised fashion to obtain a scoring function $f : \mathcal{V} \to \mathbb{R}$ that ranks abnormal nodes higher than normal ones in reverse order, i.e., $f(v_i) > f(v_j)$ for $\forall\, v_i \in \mathcal{V}_a,\ v_j \in \mathcal{V}_n$. Once trained, $f$ is applied at the inference phase to identify anomalous nodes within the same graph $\mathcal{G}$.

**Generalist GAD Setting.** Generalist GAD aims to build a universal anomaly scoring model $f$ from an collection of training graphs $\mathcal{T}_{\text{train}} = \{(\mathcal{G}_{\text{train}}^{(1)}, Y^{(1)}), \ldots, (\mathcal{G}_{\text{train}}^{(N)}, Y^{(N)})\}$. The trained $f$ is directly applicable, without fine-tuning or re-training, to unseen test graphs $\mathcal{T}_{\text{test}} = \{\mathcal{G}_{\text{test}}^{(1)}, \ldots, \mathcal{G}_{\text{test}}^{(n)}\}$

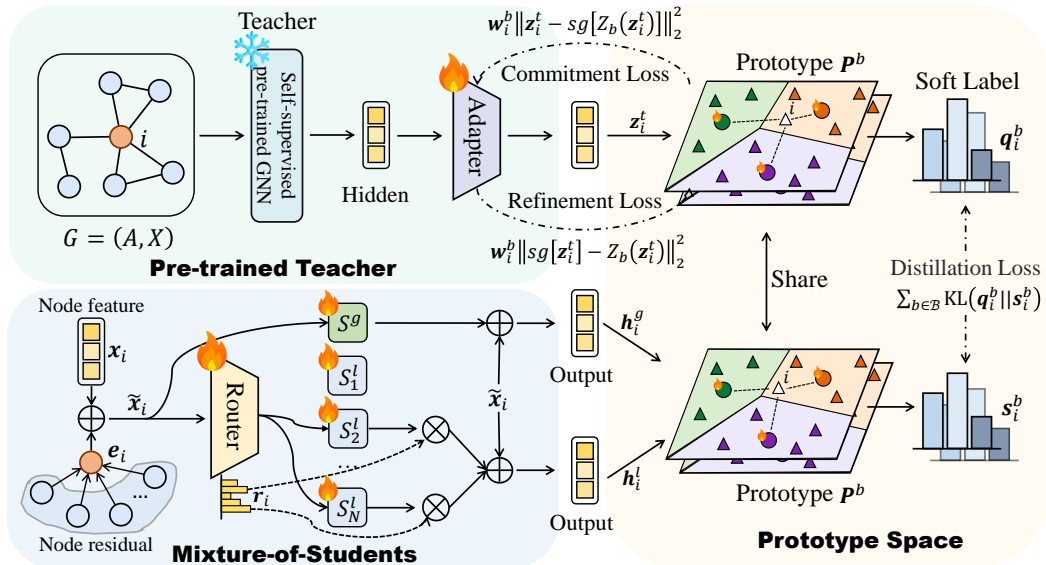

Figure 1: The architecture of the ProMoS. During training, a frozen self-supervised GNN teacher guides a Mixture-of-Students via prototype-guided soft-label distillation, while discrepancy-aware commitment and refinement objectives stabilize teacher outputs and refine the prototype for cross-graph consistency. During inference, anomalies are identified by fusing distillation bias with geometric deviation, enabling zero-shot detection on unseen graphs.

drawn from diverse domains and distributions, satisfying $\mathcal{T}_{\text{train}} \cap \mathcal{T}_{\text{test}} = \varnothing$ and even allowing for distribution differences between them. Previous generalist GAD approaches rely on labels $Y$ from training graphs $\mathcal{T}_{\text{train}}$ and even require few-shot samples at inference (Liu et al., 2024). In contrast, our study specifically emphasizes the unsupervised development of $f$, relying solely on patterns learned from $\{\mathcal{G}_{\text{train}}^{(1)}, \ldots, \mathcal{G}_{\text{train}}^{(N)}\}$ to detect anomalies in novel target graphs.

## 3.2 MIXTURE-OF-STUDENTS GUIDED BY TEACHER KNOWLEDGE

**Pre-trained Teacher.** Self-supervised GNN pretraining captures patterns of neighbor matching in unlabeled graphs, yielding robust representations that encode strong normality priors due to the dominance of normal nodes in real-world graphs (Hou et al., 2024; Zhao et al., 2025). Rather than relearning these regularities from scratch, we adopt a knowledge distillation framework that reuses a well-trained self-supervised GNN encoder as the teacher and transfers its normality priors to the student. Formally, given a graph $\mathcal{G}$, the teacher representations are computed as:

$$\mathbf{U}_T = f_T(\mathbf{X}, \mathbf{A}), \quad \mathbf{Z}_T = g_\phi(\mathbf{U}_T), \tag{1}$$

where the parameters of the teacher encoder $f_T$ are frozen. The lightweight adapter $g_\phi$ (e.g., a single-layer perceptron) is the only trainable component on the teacher side. Its parameters $\phi$ are jointly optimized with the student branches, ensuring the teacher's knowledge remains calibrated and consistent across diverse graph domains (details elaborated in Section 3.3).

**Mixture-of-Students.** To balance expressiveness and efficiency, we employ a mixture-of-students (MoS) with one *shared* and one *personalized* branch, enabling the student to capture the diverse normality patterns encoded by a frozen teacher. First, each branch operates on an enhanced node representation that augments raw features $\mathbf{x}_i$ with a residual representation $\mathbf{e}_i$, which is widely recognized to improve generalization (Qiao et al., 2025a). The resulting enhanced representation is defined as:

$$\mathbf{e}_i = \mathbf{x}_i - \frac{1}{|\mathcal{N}(v_i)|} \sum_{v_j \in \mathcal{N}(v_i)} \mathbf{x}_j, \quad \tilde{\mathbf{x}}_i = \mathbf{x}_i + \mathbf{e}_i, \tag{2}$$

where $\tilde{\mathbf{x}}_i$ is enhanced node representation. $\mathcal{N}(v_i)$ is the neighbor set of node $i$.

Given $\tilde{\mathbf{x}}_i$, the shared student $S^g$ remains always *active*, learning global normality signals from the teacher, with the shared branch formally expressed as:

$$\mathbf{h}_i^g = S^g\left(\tilde{\mathbf{x}}_i; \theta_g\right) = f_g\left(\tilde{\mathbf{x}}_i \odot \mathbf{m}_i\right) + \tilde{\mathbf{x}}_i, \tag{3}$$

where $\mathbf{h}_i^g \in \mathbb{R}^d$ is the shared student output, $f_g$ is a lightweight MLP with parameters $\theta_g$, $\odot$ denotes the Hadamard product, and $\mathbf{m}_i$ is a random binary mask for robustness (He et al., 2022). The identity skip stabilizes training and encourages $f_g$ to model teacher-induced aggregation deltas.

In contrast, the personalized branch hosts a pool of $N$ lightweight student models $\{S_p^\ell\}_{p=1}^N$. The personalized branch first employs a routing network to compute sparse activation weights, dynamically selecting a small subset of students based on masked node features to capture specialised local normality patterns. Formally, the routing network computes student selection scores as follows:

$$\mathbf{r}_i = \mathrm{softmax}\left(\mathbf{W}_r \tilde{\mathbf{x}}_i\right), \quad g_{i,p} = \begin{cases} \mathbf{r}_i[p], & \text{if } \mathbf{r}_i[p] \in \text{Top-}K(\{\mathbf{r}_i[j] | 1 \le j \le N\}), \\ 0, & \text{otherwise,} \end{cases} \tag{4}$$

where $\mathbf{r}_i \in \mathbb{R}^N$ is the routing probability vector, and $g_{i,p}$ denotes the sparse gating weight for student $p$. $\mathbf{W}_r \in \mathbb{R}^{N \times d}$ are trainable router parameters. Only the top-$K$ students receive non-zero weights, enforcing sparse activation. Given the routing weights, the personalized representation of node $v_i$ is computed as:

$$\mathbf{h}_i^\ell = S^\ell\left(\tilde{\mathbf{x}}_i; \theta_\ell\right) = \sum_{p=1}^N \left(g_{i,p} \cdot f_p(\tilde{\mathbf{x}}_i \odot \mathbf{m}_i)\right) + \tilde{\mathbf{x}}_i, \tag{5}$$

where $\mathbf{h}_i^\ell \in \mathbb{R}^d$ is the output of the personalized branch, $f_p(\cdot)$ denotes the $p$-th student parameter.

## 3.3 TRAINING OBJECTIVE

**Prototype-Driven Soft-Label Distillation.** To transfer generalizable normality priors from the frozen teacher to lightweight student branches, we introduce a prototype-driven soft-label distillation strategy. Instead of enforcing instance-level feature matching, we rely on learnable prototype codebooks that serve as semantic anchors, initialized from the node features in the training graph via k-means clustering (Douze et al., 2024) and loaded as trainable parameters. For each branch $b \in \mathcal{B} = \{g, \ell\}$ (shared $g$, personalized $\ell$), we maintain a prototype codebook $\mathbf{P}^b = [\mathbf{p}_1^b, \ldots, \mathbf{p}_{M_b}^b] \in \mathbb{R}^{M_b \times d}$. Here $M_b$ denotes the number of prototypes assigned to branch $b$. For a given branch $b$, with teacher representation $\mathbf{z}_i^t$ and the corresponding student output $\mathbf{h}_i^b$ of node $i$, we compute their distributions over the prototypes $\mathbf{P}^b$ as follows:

$$\mathbf{q}_i^b[m] = \frac{\exp\left(\mathrm{sim}(\mathbf{z}_i^t, \mathbf{p}_m^b)/\tau\right)}{\sum_{m'=1}^{M_b} \exp\left(\mathrm{sim}(\mathbf{z}_i^t, \mathbf{p}_{m'}^b)/\tau\right)}, \quad \mathbf{s}_i^b[m] = \frac{\exp\left(\mathrm{sim}(\mathbf{h}_i^b, \mathbf{p}_m^b)/\tau\right)}{\sum_{m'=1}^{M_b} \exp\left(\mathrm{sim}(\mathbf{h}_i^b, \mathbf{p}_{m'}^b)/\tau\right)}, \tag{6}$$

where $\mathbf{q}_i^b[m]$ and $\mathbf{s}_i^b[m]$ denote the probabilities of assigning node $i$ to the $m$-th prototype, produced by the teacher and the student under branch $b$, respectively. The similarity function is the negative squared Euclidean distance, i.e., $\mathrm{sim}(\mathbf{z}_i^t, \mathbf{p}_m^b) = -\left\|\mathbf{z}_i^t - \mathbf{p}_m^b\right\|_2^2$, and $\tau$ is the temperature coefficient.

Finally, the prototype-driven distillation loss minimizes the Kullback–Leibler (KL) divergence between teacher- and student-induced distributions across all branches:

$$\mathcal{L}_{\mathrm{PSD}} = \frac{1}{|\mathcal{V}|} \sum_{i=1}^{|\mathcal{V}|} \sum_{b \in \mathcal{B}} \mathrm{KL}(\mathbf{q}_i^b \,\|\, \mathbf{s}_i^b), \tag{7}$$

where $|\mathcal{V}|$ is the number of nodes in the graph and $\mathbf{q}_i^b$ are the teacher-provided soft labels.

This objective guides the students to capture prototype-level semantics distilled from the teacher, where prototypes act as high-level and abstract concepts that are easier to transfer across graphs, rather than overfitting to instance-specific details. We theoretically guarantee the effectiveness of MoS, with detailed proofs provided in Appendix A.

**Discrepancy-aware Commitment and Refinement** Graphs from different domains often exhibit substantial semantic and structural heterogeneity (Liu et al., 2024), leading to unordered or unstable

feature spaces encoded by the frozen teacher, which in turn hampers generalization across graphs. Moreover, without proper constraints, prototypes may remain largely underutilized, thereby limiting semantic coverage and diminishing representational diversity (Li et al., 2021). To mitigate these issues, we introduce two complementary objectives with distinct roles. The commitment loss regularizes the adapter outputs in Eq. 1 by pulling teacher representations toward stable prototype anchors, thereby enforcing a consistent and well-structured feature space across graphs. In contrast, the refinement loss updates the prototype to capture high-level transferable semantics. Together, these objectives ensure that the teacher features become prototype-aligned and that the prototypes evolve into meaningful semantic centers. Specifically, each teacher representation is first quantized to its nearest prototype:

$$\mathcal{Z}_b(\mathbf{z}_i^t) = \mathbf{p}_{m_i^\star}^b, \quad m_i^\star = \arg \min_{m \in [M_b]} \left\| \mathbf{z}_i^t - \mathbf{p}_m^b \right\|_2^2, \tag{8}$$

where $\mathcal{Z}_b(\mathbf{z}_i^t)$ denotes the quantized representation of node $i$ under branch $b$, $\mathbf{p}_{m_i^\star}^b$ is the $m_i$-th prototype in branch $b$.

However, directly optimizing commitment and refinement on real-world graphs is challenging because anomalous nodes inject misleading gradients, which bias prototype updates and regularizes the adapter outputs toward suboptimal alignments. To address this issue, we propose a *discrepancy-aware weighting* mechanism that adaptively downweights unreliable nodes. Specifically, we first construct the prototype–prototype relation matrix $\mathbf{Q}^b = \mathrm{softmax}\big(\mathrm{sim}(\mathbf{P}^b, \mathbf{P}^b)/\tau\big)$, which provides a global semantic structure among prototypes and serves as the ground-truth relational pattern. For each node $i$, we then measure the consistency between its teacher-induced prototype distribution $\mathbf{q}_i^b$ and the relational pattern of its nearest prototype, given by $\mathbf{Q}_{m_i^\star}^b$, which serves as a canonical reference. This consistency is quantified via the KL divergence. The core intuition is that normal nodes should yield prototype distributions well aligned with $\mathbf{Q}_{m_i^\star}^b$—since their semantics are expected to follow the same global prototype structure—resulting in low KL divergence and thus large reliability weights, whereas unreliable nodes deviate from this global prototype structure and therefore receive reduced weights. The reliability weight $w_i^b$ is computed as:

$$\tilde{w}_i^b = \sigma\Big( - \beta \cdot \big(\mathrm{KL}(\mathbf{q}_i^b \,\|\, \mathbf{Q}_{m_i^\star}^b) - \mu\big)\Big), \quad w_i^b = \frac{\tilde{w}_i^b}{\sum_{j=1}^N \tilde{w}_j^b + \epsilon}, \tag{9}$$

where $\sigma(\cdot)$ is the sigmoid function, $\beta$ controls the sharpness of reweighting, $\mu$ sets the pivot of the reliability threshold. $\beta$ and $\mu$ control the sensitivity of the reliability weight. $\mathbf{Q}_{m_i^\star}^b$ denotes the $m_i^\star$-th row of $\mathbf{Q}^b$ corresponding to the nearest prototype. Given these adaptive weights, we define the discrepancy-aware commitment and refinement loss as

$$\mathcal{L}_{\mathrm{DCR}} = \sum_{i=1}^{|\mathcal{V}|} \sum_{b \in \mathcal{B}} w_i^b \left( \left\| \mathbf{z}_i^t - \mathrm{sg}[\mathcal{Z}_b(\mathbf{z}_i^t)] \right\|_2^2 \;+\; \left\| \mathrm{sg}[\mathbf{z}_i^t] - \mathcal{Z}_b(\mathbf{z}_i^t) \right\|_2^2 \right), \tag{10}$$

where $\mathrm{sg}\,[\cdot]$ denotes the stop-gradient operation. The first term corresponds to the *commitment loss*, which pulls teacher features toward their assigned prototypes, and the second term corresponds to the *refinement loss*, which updates the prototypes so they better capture transferable semantic structure.

**Overall Objective** The overall training objective integrates prototype distillation, discrepancy-aware commitment and refinement loss:

$$\mathcal{L} = \mathcal{L}_{\mathrm{PSD}} + \lambda \, \mathcal{L}_{\mathrm{DCR}}, \tag{11}$$

where $\lambda$ are trade-off hyperparameters.

## 3.4 GENERALIST ANOMALY SCORE INFERENCE

During inference, anomaly scores are derived without retraining on the target graph. We integrate two complementary signals: (i) the distillation deviation, which reflects the semantic mismatch between teacher soft labels and student predictions, and (ii) the geometric deviation, which captures geometric inconsistency between embeddings and their quantized prototypes. The final anomaly score is a weighted combination of both terms:

$$s_i = \sum_{b \in \mathcal{B}} \left[ \mathrm{KL}\big(\mathbf{q}_i^b \,\|\, \mathbf{s}_i^b\big) + \lambda \Big( \left\| \mathbf{h}_i^b - \mathcal{Z}_b(\mathbf{h}_i^b) \right\|_2^2 + \left\| \mathbf{z}_i^t - \mathcal{Z}_b(\mathbf{z}_i^t) \right\|_2^2 \Big) \right], \tag{12}$$

where the coefficient $\lambda$ controls the relative contribution of geometric deviation. Nodes with higher $s_i$ values are considered more anomalous.

## 4 EXPERIMENTS

### 4.1 EXPERIMENTAL SETUP

**Datasets.** To evaluate the generalization ability of GAD models, we follow established protocols (Dong et al., 2024; Liu et al., 2024) by training all methods on a set of graphs and testing on a separate set of unseen graphs. Our evaluation spans 15 real-world graphs from diverse domains and scales, each containing either real or injected anomalies. Specifically, we use PubMed (Sen et al., 2008), Flickr (Tang & Liu, 2009), Questions (Platonov et al., 2023), and YelpChi (Rayana & Akoglu, 2015) as training datasets, and assess generalization on unseen in-domain graphs (Cora, CiteSeer, ACM (Tang et al., 2008), BlogCatalog (Ding et al., 2019), Facebook (Xu et al., 2022b), Weibo (Kumar et al., 2019), Reddit (Kumar et al., 2019)) and unseen out-of-domain graphs (CoAuthor CS (Shchur et al., 2018), Amazon Photo (Shchur et al., 2018), Tolokers (Likhobaba et al., 2023), T-Finance (Tang et al., 2022)). Further details are provided in Appendix D.1.

**Baselines.** We compare ProMoS against 12 representative baselines, covering both supervised and unsupervised paradigms. The supervised group includes two conventional GNNs (GCN (Kipf & Welling, 2017), GAT (Veličković et al., 2018)), three state-of-the-art GAD-specific models (BGNN (Ivanov & Prokhorenkova, 2021), BWGNN (Tang et al., 2022), GHRN (Gao et al., 2023)), as well as three recently proposed generalist GAD methods (ARC (Liu et al., 2024), UNPrompt (Niu et al., 2025), AnomalyGFM (Qiao et al., 2025a)). The unsupervised group covers four representative paradigms: the reconstruction-based method DOMINANT (Ding et al., 2019), the contrastive method CoLA (Liu et al., 2021), the hop-prediction method HCM-A (Huang et al., 2022), and the affinity-based method TAM (Qiao & Pang, 2023).

**Implementation.** Following prior GAD protocols (Ding et al., 2019; Liu et al., 2021; Qiao et al., 2025a), we report AUROC and AUPRC (mean±std over five runs with different seeds). To enable a fair assessment of generalist GAD, we focus on zero-shot inference: train on training graphs $\mathcal{T}_{\text{train}}$ and evaluate on unseen test graphs $\mathcal{T}_{\text{test}}$ with no support set. Comparisons to few-shot methods (e.g., ARC (Liu et al., 2024)) are deferred to Appendix E.2. For feature parity, we apply the projection mapping of (Liu et al., 2024) to obtain 64-dimensional node features. Our pre-trained teacher GNN uses GCA (Zhu et al., 2021), implemented via the PyG-SSL Toolkit (Zheng et al., 2024). All baselines are implemented via official code and tuned following their reported strategies. More implementation details are provided in Appendix D.2.

### 4.2 GENERALIST GAD PERFORMANCE

We evaluate the generalist GAD performance across eleven datasets from diverse domains. Table 2 reports the AUROC comparison with existing supervised and unsupervised baselines, while detailed AUPRC results are provided in Appendix E.1. Several observations emerge.

First, supervised pre-training methods struggle in this setting. This highlights the inherent diversity of anomalies, and models that emphasize learning specific abnormal patterns from training graphs have difficulty generalizing to unseen anomaly types. Second, unsupervised GAD methods, such as TAM, perform more robustly by modeling normality to identify outliers, confirming the promise of this direction. However, they still suffer from substantial degradation in the generalist setting. For instance, CoLA achieves AUROC scores (%) of 87.79 and 89.68 on Cora and CiteSeer, respectively, when training and testing on the same graph, yet loses over 24% when generalized across domains.

Finally, ours consistently outperforms both supervised and unsupervised baselines while requiring only unsupervised pre-training. Among the eleven datasets, ours ranks first on nine and second on one, achieving an average AUROC improvement of 14.12% over the strongest baseline DOMINANT. These gains stem from leveraging advances in graph self-supervised learning and our tailored design mixture-of-students, prototype distillation, and discrepancy-aware objectives, which enable more faithful modeling of generalizable normal patterns for anomaly detection.

Table 2: Anomaly detection performance on eleven datasets under the zero-shot setting. We report the mean and standard deviation of AUROC. $1^{st}$ marks the best result, $2^{nd}$ the runner-up, and $3^{rd}$. *OOM* denotes out-of-memory.

| Method | Cora | CiteSeer | ACM | BlogCatalog | Facebook | Weibo |
|---|---|---|---|---|---|---|
| Supervised - Pre-Train Only | | | | | | |
| GCN | $59.64_{\pm 8.30}$ | $60.27_{\pm 8.11}$ | $60.49_{\pm 9.65}$ | $56.19_{\pm 6.39}$ | $29.51_{\pm 4.86}$ | $76.64_{\pm 17.69}$ |
| GAT | $50.06_{\pm 2.65}$ | $51.59_{\pm 3.49}$ | $48.79_{\pm 2.73}$ | $50.40_{\pm 2.80}$ | $51.88_{\pm 2.16}$ | $53.06_{\pm 7.48}$ |
| BGNN | $42.45_{\pm 11.57}$ | $42.32_{\pm 11.82}$ | $44.00_{\pm 13.69}$ | $47.67_{\pm 8.52}$ | $54.74_{\pm 25.29}$ | $32.75_{\pm 35.35}$ |
| BWGNN | $54.06_{\pm 3.27}$ | $52.61_{\pm 2.88}$ | $67.59_{\pm 0.70}$ | $56.34_{\pm 1.21}$ | $45.84_{\pm 4.97}$ | $53.38_{\pm 1.61}$ |
| GHRN | $59.89_{\pm 6.57}$ | $56.04_{\pm 9.19}$ | $55.65_{\pm 6.37}$ | $57.64_{\pm 3.48}$ | $44.81_{\pm 8.06}$ | $51.87_{\pm 14.18}$ |
| UNPrompt | $53.19_{\pm 4.12}$ | $53.70_{\pm 4.08}$ | $68.74_{\pm 0.88}$ | $68.87_{\pm 0.56}$ | $61.37_{\pm 4.54}$ | $44.94_{\pm 4.73}$ |
| AnomalyGFM | $47.83_{\pm 0.54}$ | $49.10_{\pm 1.36}$ | $53.40_{\pm 1.09}$ | $49.31_{\pm 1.63}$ | $56.55_{\pm 1.98}$ | $51.24_{\pm 0.41}$ |
| Unsupervised - Pre-Train Only | | | | | | |
| DOMINANT | $66.53_{\pm 1.15}$ | $69.47_{\pm 2.02}$ | $70.08_{\pm 2.34}$ | $74.25_{\pm 0.65}$ | $51.01_{\pm 0.78}$ | $92.88_{\pm 0.32}$ |
| CoLA | $63.29_{\pm 8.88}$ | $62.84_{\pm 9.52}$ | $66.85_{\pm 4.43}$ | $50.04_{\pm 3.25}$ | $12.99_{\pm 11.68}$ | $16.27_{\pm 5.64}$ |
| HCM-A | $54.28_{\pm 4.73}$ | $48.12_{\pm 6.80}$ | $53.70_{\pm 4.64}$ | $55.31_{\pm 0.57}$ | $35.44_{\pm 13.97}$ | $65.52_{\pm 12.58}$ |
| TAM | $62.02_{\pm 2.39}$ | $72.27_{\pm 0.83}$ | $74.43_{\pm 1.59}$ | $49.86_{\pm 0.73}$ | $65.88_{\pm 6.66}$ | $71.54_{\pm 0.18}$ |
| Unsupervised - Pre-Train Only | | | | | | |
| Ours | $84.56_{\pm 0.16}$ | $90.77_{\pm 0.12}$ | $89.47_{\pm 0.79}$ | $76.17_{\pm 0.37}$ | $69.31_{\pm 0.50}$ | $91.74_{\pm 0.03}$ |

| Method | Reddit | CS | Photo | Tolokers | T-Finance | Avg. |
|---|---|---|---|---|---|---|
| Supervised - Pre-Train Only | | | | | | |
| GCN | $50.43_{\pm 4.41}$ | $47.90_{\pm 2.12}$ | $52.65_{\pm 3.38}$ | $50.79_{\pm 0.32}$ | $64.37_{\pm 0.51}$ | $55.35_{\pm 11.75}$ |
| GAT | $51.78_{\pm 4.04}$ | $49.02_{\pm 0.53}$ | $51.71_{\pm 1.00}$ | $54.86_{\pm 2.24}$ | $65.56_{\pm 0.07}$ | $52.61_{\pm 4.64}$ |
| BGNN | $50.27_{\pm 3.84}$ | $52.35_{\pm 4.63}$ | $49.88_{\pm 1.13}$ | $51.53_{\pm 4.29}$ | $50.78_{\pm 5.46}$ | $47.16_{\pm 6.30}$ |
| BWGNN | $48.97_{\pm 5.74}$ | $57.09_{\pm 5.85}$ | $51.29_{\pm 3.88}$ | $53.82_{\pm 2.58}$ | $52.88_{\pm 4.56}$ | $53.99_{\pm 5.50}$ |
| GHRN | $46.22_{\pm 2.33}$ | $59.49_{\pm 9.06}$ | $54.24_{\pm 8.95}$ | $48.29_{\pm 9.02}$ | $52.12_{\pm 16.68}$ | $53.30_{\pm 5.15}$ |
| UNPrompt | $57.10_{\pm 1.21}$ | $71.64_{\pm 0.86}$ | $52.88_{\pm 3.67}$ | $38.60_{\pm 2.85}$ | $22.14_{\pm 2.75}$ | $53.92_{\pm 14.66}$ |
| AnomalyGFM | $52.78_{\pm 0.88}$ | $48.24_{\pm 0.98}$ | $49.65_{\pm 0.79}$ | $48.16_{\pm 0.87}$ | $64.44_{\pm 7.12}$ | $51.88_{\pm 4.96}$ |
| Unsupervised - Pre-Train Only | | | | | | |
| DOMINANT | $50.05_{\pm 4.92}$ | $60.61_{\pm 0.11}$ | $47.39_{\pm 0.34}$ | $48.12_{\pm 1.65}$ | *OOM* | $63.04_{\pm 14.55}$ |
| CoLA | $52.81_{\pm 6.69}$ | $52.72_{\pm 1.02}$ | $52.84_{\pm 1.95}$ | $51.02_{\pm 2.81}$ | $52.61_{\pm 3.51}$ | $48.57_{\pm 17.72}$ |
| HCM-A | $48.79_{\pm 2.75}$ | $62.91_{\pm 5.86}$ | $50.62_{\pm 1.40}$ | $54.46_{\pm 0.92}$ | *OOM* | $52.92_{\pm 8.30}$ |
| TAM | $55.43_{\pm 0.33}$ | $69.95_{\pm 0.12}$ | $58.35_{\pm 0.21}$ | $50.51_{\pm 0.14}$ | $56.16_{\pm 4.54}$ | $62.40_{\pm 8.93}$ |
| Unsupervised - Pre-Train Only | | | | | | |
| Ours | $60.83_{\pm 0.35}$ | $88.85_{\pm 0.60}$ | $72.67_{\pm 1.07}$ | $52.80_{\pm 0.82}$ | $71.62_{\pm 1.06}$ | $77.16_{\pm 13.09}$ |

## 4.3 ABLATION STUDY

To comprehensively assess the contribution of each component in ProMoS, we construct six variants: (1) w/o PSD removes the prototype-driven soft-label distillation objective; (2) w/o DCR discards the discrepancy-aware commitment and refinement loss; (3) w/o SSL replaces the pre-trained teacher GNN with a randomly initialized two-layer GCN; (4) w/o PB drops the personalized branch in the MoS design; (5) w/o SB drops the shared branch; and (6) w/o DIS removes the quality control weights $w_i^b$ in the commitment and refinement stage. Table 3 reports the results. We observe that eliminating any single module consistently harms performance. In particular, removing the prototype-driven distillation mechanism (w/o PSD) leads to a dramatic drop, with performance on most datasets barely above random guessing, highlighting the necessity of prototype-guided distillation. Similarly, excluding the constraint on teacher outputs (w/o DCR) or discarding the pre-trained teacher in favor of learning from scratch (w/o SSL) results in more than 5% AUROC reduction. Comparing MoS branches, w/o SB performs better than w/o PB, suggesting the importance of the personalized branch for capturing complementary knowledge. Finally, disabling the data-quality weighting in the commitment and refinement stage (w/o DIS) also causes a measurable decline, further confirming its role in stabilizing training and improving detection robustness.

Table 3: Ablation results w.r.t. AUC for ProMoS and its variants.

| Method | ACM | Facebook | Reddit | CS | Photo | T-Finance | Avg. |
|---|---|---|---|---|---|---|---|
| ProMoS | $89.47_{\pm0.79}$ | $69.31_{\pm0.50}$ | $60.83_{\pm0.35}$ | $88.85_{\pm0.60}$ | $72.67_{\pm1.07}$ | $71.62_{\pm1.06}$ | $75.46$ |
| w/o PSD | $69.25_{\pm0.42}$ | $26.68_{\pm1.09}$ | $56.72_{\pm1.95}$ | $65.75_{\pm0.34}$ | $55.91_{\pm1.03}$ | $63.58_{\pm0.66}$ | $56.32$ |
| w/o DCR | $88.11_{\pm0.93}$ | $65.86_{\pm1.05}$ | $56.14_{\pm1.02}$ | $87.23_{\pm0.65}$ | $58.80_{\pm5.80}$ | $65.05_{\pm3.41}$ | $70.20$ |
| w/o SSL | $76.60_{\pm0.69}$ | $68.93_{\pm0.35}$ | $53.83_{\pm1.82}$ | $79.27_{\pm0.34}$ | $71.01_{\pm1.14}$ | *OOM* | $69.93$ |
| w/o PB | $89.39_{\pm1.01}$ | $68.63_{\pm0.59}$ | $59.78_{\pm1.80}$ | $88.33_{\pm0.25}$ | $69.49_{\pm5.61}$ | $69.88_{\pm1.33}$ | $74.25$ |
| w/o SB | $89.21_{\pm0.93}$ | $69.16_{\pm0.25}$ | $58.48_{\pm2.89}$ | $88.27_{\pm1.66}$ | $72.09_{\pm1.29}$ | $70.63_{\pm0.48}$ | $74.64$ |
| w/o DIS | $88.99_{\pm0.88}$ | $68.81_{\pm0.18}$ | $59.55_{\pm1.87}$ | $88.12_{\pm0.33}$ | $72.17_{\pm0.78}$ | $69.49_{\pm1.19}$ | $74.52$ |

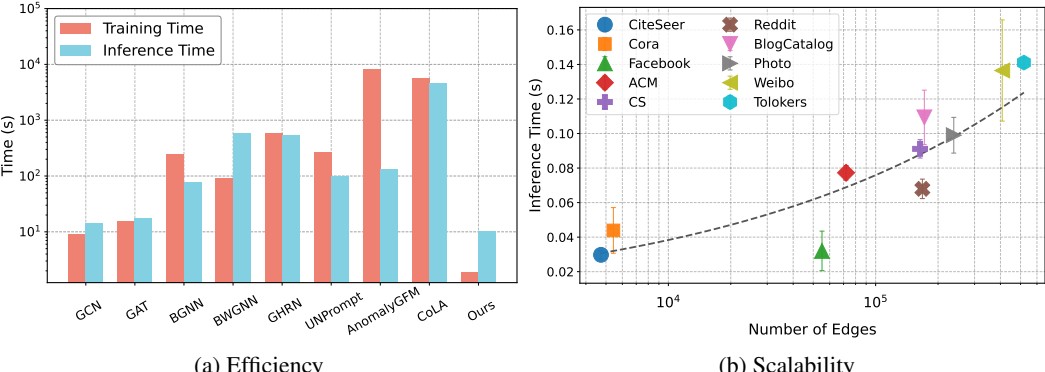

(a) Efficiency  (b) Scalability

Figure 2: Efficiency and scalability. (a) Training and inference time (seconds, log-scale) across baselines; our method achieves the lowest overall cost. (b) Inference time as a function of edge count (log). The dashed curve shows a power-law fit $T \propto |\mathcal{E}|^\alpha$ with $\alpha \approx 0.3$, indicating sub-linear growth ($\alpha \approx 1$ would be near-linear).

## 4.4 Efficiency and Scalability Analysis

**Efficiency.** First, we measure total training time on the training graphs $\mathcal{T}_{\text{train}}$ and inference time on unseen test graphs $\mathcal{T}_{\text{test}}$, with each epoch training efficiency and theoretical time complexity analysis provided in Appendix C. As shown in Fig. 2a, GCN and GAT are the fastest baselines due to their minimalist architectures, while CoLA is extremely slow because it performs many random walks during both training and inference (e.g., 256 walks per node at inference). ProMoS attains the best overall efficiency, running 4.8× faster than GCN during training and 1.4× faster during inference.

**Scalability.** Second, we measure inference time on ten graphs (4,732-519,000 edges), reporting mean±std over five runs (x-axis in log-scale). As shown in Fig. 2b, the results follow a power-law trend $T \propto |\mathcal{E}|^\alpha$ with $\alpha \approx 0.3$. Since the slope is well below 1, inference grows *sub-linearly* with graph size: a tenfold increase in edge size leads to only about a twofold increase in inference time. This result demonstrates that ProMoS scales efficiently to large graphs. To further validate ProMoS's adaptability to different teachers, we replace the teacher with GCA (default) (Zhu et al., 2021), GraphCL (You et al., 2020), BGRL (Thakoor et al., 2022), DGI (Veličković et al., 2019), and GraphMAE (Hou et al., 2022). As shown in Table 4, performance remains comparable with low variance across all choices, underscoring robustness to teacher selection and ProMoS's plug-and-play design (more in Appendix E.3).

## 5 Conclusion

In this work, we introduced ProMoS, the first fully unsupervised generalist GAD framework that enables zero-shot detection on unseen graphs. ProMoS transfers rich normal patterns from a frozen self-supervised GNN teacher to a lightweight mixture-of-students via prototype-guided soft-label distillation in a learnable high-level semantic space. Our discrepancy-aware commitment and re-

Table 4: AUROC of ProMoS with different pre-trained teacher backbones.

| Method | Cora | CiteSeer | ACM | BlogCatalog | Weibo | Tolokers | Avg. |
|---|---|---|---|---|---|---|---|
| ProMoS + GCA | $84.56_{\pm 0.16}$ | $90.77_{\pm 0.12}$ | $\mathbf{89.47}_{\pm 0.79}$ | $76.17_{\pm 0.37}$ | $\mathbf{91.74}_{\pm 0.03}$ | $52.80_{\pm 0.82}$ | $\mathbf{80.92}$ |
| ProMoS + GraphCL | $\mathbf{89.20}_{\pm 0.93}$ | $\mathbf{93.42}_{\pm 0.27}$ | $81.32_{\pm 0.58}$ | $66.80_{\pm 0.81}$ | $91.30_{\pm 0.06}$ | $54.66_{\pm 0.23}$ | $79.45$ |
| ProMoS + BGRL | $87.56_{\pm 0.20}$ | $89.60_{\pm 0.38}$ | $73.97_{\pm 0.34}$ | $\mathbf{76.96}_{\pm 0.11}$ | $91.55_{\pm 0.03}$ | $\mathbf{56.69}_{\pm 0.25}$ | $79.39$ |
| ProMoS + DGI | $84.62_{\pm 0.02}$ | $90.90_{\pm 0.02}$ | $84.97_{\pm 0.77}$ | $76.63_{\pm 0.21}$ | $91.66_{\pm 0.02}$ | $51.30_{\pm 1.44}$ | $80.01$ |
| ProMoS + GraphMAE | $83.76_{\pm 0.09}$ | $90.51_{\pm 0.09}$ | $76.83_{\pm 0.47}$ | $73.76_{\pm 0.34}$ | $91.73_{\pm 0.07}$ | $50.24_{\pm 0.64}$ | $77.81$ |

finement mechanism further enhances generalization by stabilizing teacher outputs and refining semantic prototypes. Extensive experiments across 11 real-world graphs demonstrate that ProMoS consistently outperforms state-of-the-art baselines in both accuracy and efficiency. Overall, this work lays the groundwork for scalable, label-free generalist graph anomaly detection.

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

# A  PROOF OF THEOREM 1

**Theorem 1** (Error reduction of MoS). *Consider a graph $\mathcal{G} = (\mathcal{V}, \mathcal{E}, \mathbf{X})$. For any node $i \in \mathcal{V}$, let the Mixture-of-Students (MoS) architecture consist of $\{S_k\}_{k=1}^N$ student models with outputs $\widehat{f}(i)$. Then, for any single student $k \in \{1, \ldots, N\}$, the MoS prediction achieves no larger expected prediction error than the single student:*

$$\mathbb{E}\Big[(y_i - \widehat{f}(i))^2 \,\Big|\, \mathcal{G}\Big] \;\leq\; \mathbb{E}\big[(y_i - f_k(i))^2 \,|\, \mathcal{G}\big]. \tag{13}$$

where $y_i$ denotes the ground-truth label of node $i$, $f_k(i)$ is the output of the $k$-th student.

*Proof.* To formalize the predictive mechanism of the Mixture-of-Students (MoS), we describe how multiple student models are combined through router weights to produce the final output. Specifically, the MoS prediction for node $i \in \mathcal{V}$ is defined as the weighted sum:

$$\widehat{f}(i) \;=\; \sum_{k=1}^N g_{i,k}\, f_k(i) \;=\; \mathbf{g}_i^\top \mathbf{f}(i), \tag{14}$$

where $f_k(i)$ denotes the output of the $k$-th student $S_k$, $\mathbf{g}_i = (g_{i,1}, \ldots, g_{i,N})^\top \in \Delta^N$ are the router weights with $\sum_{k=1}^N g_{i,k} = 1$ and $g_{i,k} \geq 0$, and $\mathbf{f}(i) = (f_1(i), \ldots, f_N(i))^\top$ stacks all individual student outputs.

To analyze the prediction error of MoS, we expand the expected squared loss and separate it into bias, variance, and noise. Each label is modeled as $y_i = f_i^\star + \epsilon$, where $f_i^\star$ represents the deterministic ground-truth component of node $i$, and $\epsilon$ is a stochastic residual typically modeled as Gaussian noise $\epsilon \sim \mathcal{N}(0, \sigma^2)$. Under this formulation, the decomposition proceeds as:

$$
\begin{aligned}
\mathbb{E}\Big[(y_i - \widehat{f}(i))^2\Big] &= \mathbb{E}\Big[y_i^2 - 2y_i\widehat{f}(i) + \widehat{f}(i)^2 \mid \mathcal{G}\Big] \\[4pt]
&= \mathbb{E}[y_i^2] - 2\mathbb{E}[y_i\widehat{f}(i)] + \mathbb{E}[\widehat{f}(i)^2] \\[4pt]
&= \mathbb{E}[(f_i^\star + \epsilon)^2] - 2\mathbb{E}[(f_i^\star + \epsilon)\widehat{f}(i)] + \mathbb{E}[\widehat{f}(i)^2] \\[4pt]
&= \mathbb{E}[(f_i^\star)^2] + 2\mathbb{E}[f_i^\star\epsilon] + \mathbb{E}[\epsilon^2] - 2\mathbb{E}[f_i^\star\widehat{f}(i)] - 2\mathbb{E}[\epsilon\widehat{f}(i)] + \mathbb{E}[\widehat{f}(i)^2] \\[4pt]
&= (f_i^\star)^2 + \sigma^2 - 2f_i^\star\mathbb{E}[\widehat{f}(i)] - 2\mathbb{E}[\epsilon]\mathbb{E}[\widehat{f}(i)] + \mathbb{E}[\widehat{f}(i)^2] \\[4pt]
&= (f_i^\star)^2 + \sigma^2 - 2f_i^\star\,\mathbb{E}[\widehat{f}(i)] + \mathbb{E}[\widehat{f}(i)^2] \\[4pt]
&= (f_i^\star)^2 + \sigma^2 - 2f_i^\star\,\mathbb{E}[\widehat{f}(i)] + \mathrm{Var}(\widehat{f}(i)) + \big(\mathbb{E}[\widehat{f}(i)]\big)^2 \\[4pt]
&= \underbrace{\big(f_i^\star - \mathbb{E}[\widehat{f}(i) \mid \mathcal{G}]\big)^2}_{\text{Bias}} + \underbrace{\mathrm{Var}(\widehat{f}(i) \mid \mathcal{G})}_{\text{Variance}} + \underbrace{\sigma^2(\mathcal{G})}_{\text{Noise}}.
\end{aligned}
\tag{15}
$$

In parallel, the expected squared loss of a single student $S_k$ can be decomposed in the same manner, yielding:

$$\mathbb{E}\big[(y_i - f_k(i))^2 \,\big|\, \mathcal{G}\big] = \underbrace{\big(f_i^\star - \mathbb{E}[f_k(i) \mid \mathcal{G}]\big)^2}_{\text{Bias}} + \underbrace{\mathrm{Var}(f_k(i) \mid \mathcal{G})}_{\text{Variance}} + \underbrace{\sigma^2(\mathcal{G})}_{\text{Noise}}. \tag{16}$$

To establish a direct comparison between MoS and an individual student $S_k$, we consider the difference in their conditional prediction errors, $\mathbb{E}\Big[(y_i - \widehat{f}(i))^2 \mid \mathcal{G}\Big] - \mathbb{E}\big[(y_i - f_k(i))^2 \mid \mathcal{G}\big]$. Substituting the bias–variance–noise decomposition into both terms gives:

$$\mathbb{E}\Big[(y_i - \widehat{f}(i))^2 \,\Big|\, \mathcal{G}\Big] - \mathbb{E}\big[(y_i - f_k(i))^2 \,\big|\, \mathcal{G}\big]$$
$$= \underbrace{\Big(f_i^\star - \mathbb{E}[\widehat{f}(i) \mid \mathcal{G}]\Big)^2 - \Big(f_i^\star - \mathbb{E}[f_k(i) \mid \mathcal{G}]\Big)^2}_{\text{Bias difference}} + \underbrace{\mathrm{Var}(\widehat{f}(i) \mid \mathcal{G}) - \mathrm{Var}(f_k(i) \mid \mathcal{G})}_{\text{Variance difference}}. \tag{17}$$

Then, we introduce a mild mean alignment hypothesis reflecting that all students are trained under the same target/teacher and differ only in stochasticity:

$$\mathbb{E}[f_k(i) \mid \mathcal{G}] = \mu_i(\mathcal{G}) \qquad \text{for all } k \in \{1, \ldots, N\}. \tag{18}$$

Combining Eq. 18 with the MoS predictor in Eq. 14 and the simplex constraint $\mathbf{1}^\top \mathbf{g}_i = 1$, we obtain

$$\mathbb{E}\Big[\widehat{f}(i) \mid \mathcal{G}\Big] = \mathbb{E}\big[\mathbf{g}_i^\top \mathbf{f}(i) \mid \mathcal{G}\big] = \mathbf{g}_i^\top \mathbb{E}[\mathbf{f}(i) \mid \mathcal{G}] = \mathbf{g}_i^\top \big(\mu_i(\mathcal{G})\mathbf{1}\big) = \mu_i(\mathcal{G}). \tag{19}$$

Therefore the two bias terms in Eq. 17 coincide:

$$\Big(f_i^\star - \mathbb{E}[\widehat{f}(i) \mid \mathcal{G}]\Big)^2 = \Big(f_i^\star - \mathbb{E}[f_k(i) \mid \mathcal{G}]\Big)^2 = \big(f_i^\star - \mu_i(\mathcal{G})\big)^2,$$

and the bias difference vanishes. Consequently, the expected prediction error simplifies to the:

$$\mathbb{E}\Big[(y_i - \widehat{f}(i))^2 \mid \mathcal{G}\Big] - \mathbb{E}\big[(y_i - f_k(i))^2 \mid \mathcal{G}\big] = \mathrm{Var}(\widehat{f}(i) \mid \mathcal{G}) - \mathrm{Var}(f_k(i) \mid \mathcal{G}). \tag{20}$$

Having reduced the comparison to the variance gap, we now compute both terms explicitly. Let $\mathbf{m} := \mathbb{E}[\mathbf{f}(i) \mid \mathcal{G}]$, and $\Sigma_\mathcal{G} := \mathrm{Cov}(\mathbf{f}(i) \mid \mathcal{G}) = \mathbb{E}\big[(\mathbf{f} - \mathbf{m})(\mathbf{f} - \mathbf{m})^\top \mid \mathcal{G}\big]$. Since the router weights $\mathbf{g}_i$ are $\mathcal{G}$–measurable (given $\mathcal{G}$ and node $i$), the conditional variance of the MoS predictor satisfies:

$$\begin{aligned}
\mathrm{Var}(\widehat{f}(i) \mid \mathcal{G}) &= \mathbb{E}\Big[\big(\widehat{f}(i) - \mathbb{E}[\widehat{f}(i) \mid \mathcal{G}]\big)^2 \,\Big|\, \mathcal{G}\Big] \\
&= \mathbb{E}\Big[\big(\mathbf{g}_i^\top \mathbf{f} - \mathbf{g}_i^\top \mathbf{m}\big)^2 \,\Big|\, \mathcal{G}\Big] \\
&= \mathbb{E}\Big[\big(\mathbf{g}_i^\top (\mathbf{f} - \mathbf{m})\big)^2 \,\Big|\, \mathcal{G}\Big] \\
&= \mathbb{E}\big[(\mathbf{f} - \mathbf{m})^\top (\mathbf{g}_i \mathbf{g}_i^\top)(\mathbf{f} - \mathbf{m}) \mid \mathcal{G}\big] \\
&= \mathrm{tr}\big(\mathbf{g}_i \mathbf{g}_i^\top \, \mathbb{E}\big[(\mathbf{f} - \mathbf{m})(\mathbf{f} - \mathbf{m})^\top \mid \mathcal{G}\big]\big) \\
&= \mathrm{tr}\big(\mathbf{g}_i \mathbf{g}_i^\top \Sigma_\mathcal{G}\big) = \mathbf{g}_i^\top \Sigma_\mathcal{G} \, \mathbf{g}_i.
\end{aligned} \tag{21}$$

To further simplify the variance expression, we impose a standard *equicorrelation* structure on the student outputs. Specifically, we assume that each student has the same conditional variance $v(\mathcal{G})$ and any pair shares a common conditional correlation $\rho(\mathcal{G}) \in [-1, 1]$. Formally,

$$\mathrm{Var}(f_k(i) \mid \mathcal{G}) = v(\mathcal{G}), \quad \mathrm{Corr}(f_k(i), f_r(i) \mid \mathcal{G}) = \rho(\mathcal{G}), \ \forall k \neq r. \tag{22}$$

Under this assumption, the covariance matrix admits the closed form:

$$\Sigma_\mathcal{G} = v(\mathcal{G})\Big((1 - \rho(\mathcal{G}))I + \rho(\mathcal{G}) \, \mathbf{1}\mathbf{1}^\top\Big), \tag{23}$$

where $\mathbf{1}$ is the all-ones vector in $\mathbb{R}^N$.

We next compute $\mathrm{Var}(\widehat{f}(i) \mid \mathcal{G})$ under the Eq. 21 that $\mathrm{Var}(\widehat{f}(i) \mid \mathcal{G}) = \mathbf{g}_i^\top \Sigma_{\mathcal{G}}\, \mathbf{g}_i$, and from Eq. 23 that $\Sigma_{\mathcal{G}} = v(\mathcal{G})\big((1 - \rho(\mathcal{G}))I + \rho(\mathcal{G})\mathbf{1}\mathbf{1}^\top\big)$. Substituting and simplifying yields:

$$
\begin{aligned}
\mathrm{Var}(\widehat{f}(i) \mid \mathcal{G}) &= \mathbf{g}_i^\top \Big[ v(\mathcal{G})\big((1 - \rho(\mathcal{G}))I + \rho(\mathcal{G})\mathbf{1}\mathbf{1}^\top\big) \Big] \mathbf{g}_i \\
&= v(\mathcal{G})\Big( (1 - \rho(\mathcal{G}))\, \mathbf{g}_i^\top \mathbf{g}_i \ + \ \rho(\mathcal{G})\, \mathbf{g}_i^\top \mathbf{1}\mathbf{1}^\top \mathbf{g}_i \Big) \\
&= v(\mathcal{G})\Big( (1 - \rho(\mathcal{G}))\, \|\mathbf{g}_i\|_2^2 \ + \ \rho(\mathcal{G})\, (\mathbf{1}^\top \mathbf{g}_i)^2 \Big) \\
&= v(\mathcal{G})\Big( \rho(\mathcal{G}) + (1 - \rho(\mathcal{G}))\|\mathbf{g}_i\|_2^2 \Big).
\end{aligned}
\tag{24}
$$

Therefore, combining the decompositions, we have:

$$
\begin{aligned}
\mathbb{E}\Big[(y_i - \widehat{f}(i))^2 \mid \mathcal{G}\Big] - \mathbb{E}\big[(y_i - f_k(i))^2 \mid \mathcal{G}\big] &= \mathrm{Var}(\widehat{f}(i) \mid \mathcal{G}) - \mathrm{Var}(f_k(i) \mid \mathcal{G}) \\
&= v(\mathcal{G})\Big( \rho(\mathcal{G}) + (1 - \rho(\mathcal{G}))\|\mathbf{g}_i\|_2^2 \Big) - v(\mathcal{G}) \\
&= -v(\mathcal{G})\big(1 - \rho(\mathcal{G})\big)\big(1 - \|\mathbf{g}_i\|_2^2\big) \ \leq \ 0.
\end{aligned}
\tag{25}
$$

*where* $v(\mathcal{G}) \geq 0$, $\rho(\mathcal{G}) \leq 1$, and $\|\mathbf{g}_i\|_2^2 \leq 1$ for any $\mathbf{g}_i \in \Delta^N$ (equality iff $\mathbf{g}_i$ is one–hot).

In particular, if the router activates at least two students, then $\|\mathbf{g}_i\|_2^2 < 1$ and the inequality is strict. This completes the proof. $\qquad\square$

## B  ALGORITHMS

Algorithm 1 outlines ProMoS, prototype-guided mixture-of-students framework for generalist GAD.

**Initialization.** We freeze the pretrained teacher $f_T$, introduce an adapter $g_\phi$, and initialize a shared student $S_g$, personalized students $S_{p\,p=1}^{\ell\,N}$, router $\mathbf{W}_r$, and prototypes $\mathbf{P}_b$ from clustered node embeddings.

**Training.** Given training graphs, the teacher produces calibrated features. The shared branch models global patterns, while the personalized branch employs sparse Top-$K$ routing for efficiency. Students are guided by prototype soft-label distillation, enforcing alignment with teacher semantics, and by discrepancy-aware commitment and refinement, which stabilize the teacher's output and update prototypes using sample reliability weighting. The total loss combines these two objectives.

**Inference.** On unseen graphs, no retraining is required. We compute teacher and student features, project them into prototype space, and score anomalies by combining distillation bias with geometric deviation. This enables zero-shot detection across diverse graphs.

## C  TIME COMPLEXITY ANALYSIS

**Theoretical Analysis.** In this section, we analyze the time complexity of ProMoS by dividing it into the encoder and the loss functions. Since the teacher model is pre-computed offline, its cost is negligible and omitted from the analysis. For the encoder, the main components include the adapter, the router, and the student models. The adapter projects each node representation into the student space with complexity $\mathcal{O}(nd^2)$, where $n = |\mathcal{V}|$ is the number of nodes and $d$ is the embedding dimension. The router computes gating logits over $N$ students and performs a Top-$K$ selection, leading to $\mathcal{O}(ndN)$. For student forward propagation, one shared student and $K$ activated personalized students are applied to each node, giving $\mathcal{O}(n(K+1)d^2)$. For the loss functions, the prototype-driven soft-label distillation requires computing similarities between each node and $M$ prototypes, which takes $\mathcal{O}(ndM)$, where $M$ is the number of prototypes. The commitment loss and refinement loss introduce an additional residual computation with $\mathcal{O}(nd)$ and prototype–prototype similarity construction with $\mathcal{O}(dM^2)$. Finally, router regularization terms such as load-balancing

---

**Algorithm 1:** ProMoS: Prototype-Guided Mixture-of-Students for Generalist GAD

---

**Input** : Training graphs $\mathcal{T}_{\text{train}}$; frozen teacher $f_T$; adapter $g_\phi$; shared student $S_g$; personalized students $\{S_p^\ell\}_{p=1}^N$; prototype codebooks $\{\mathbf{P}_b\}_{b\in\{g,\ell\}}$; epochs $E$; temperature $\tau$; Top-$K$ router.

**Output:** Zero-shot anomaly scoring function $s : V \to \mathbb{R}$.

1 **Stage A: Initialization**

2 Initialize learnable parameters $\theta_g$, $\{\theta_p^\ell\}_{p=1}^N$, router $\mathbf{W}_r$, adapter $\phi$

3 Initialize prototypes $\{\mathbf{P}_b\}$ by clustering node embeddings (e.g., k-means)

4 **Stage B: Training (teacher frozen)**

5 **for** $e = 1$ **to** $E$ **do**

6     **for** $\mathcal{D}^{(i)} \in \mathcal{T}_{train}$ **do**

7         Extract node features and adjacency $(\mathbf{X}^{(i)}, \mathbf{A}^{(i)})$

        `// Teacher forward (stop-grad) and calibration`

8         $\mathbf{U}_T \leftarrow f_T(\mathbf{X}^{(i)}, \mathbf{A}^{(i)}); \quad \mathbf{Z}_T \leftarrow g_\phi(\mathbf{U}_T)$

        `// Shared branch`

9         $\mathbf{h}^g \leftarrow S_g\left(\tilde{\mathbf{X}}_i^{(i)}; \theta_g\right)$

        `// Personalized branch with sparse Top-K routing`

10        $\mathbf{r} \leftarrow \text{softmax}(\mathbf{W}_r \tilde{\mathbf{X}}_i^{(i)}); \quad g_{i,p} \leftarrow \text{Top-}K(\mathbf{r}_i); \quad \mathbf{h}^\ell \leftarrow \sum_{p=1}^N g_{i,p} S_p^\ell\left(\tilde{\mathbf{X}}_i^{(i)}; \theta_p^\ell\right)$

        `// Prototype-guided soft-label distillation (PSD)`

11        **for** $b \in \{g, \ell\}$ **do**

12           $\mathbf{q}_i^b \propto \exp(\text{sim}(\mathbf{z}_i^t, \mathbf{P}_b)/\tau); \quad \mathbf{s}_i^b \propto \exp(\text{sim}(\mathbf{h}_i^b, \mathbf{P}_b)/\tau)$

13        $\mathcal{L}_{\text{PSD}} \leftarrow \dfrac{1}{|V|} \sum_i \sum_b \text{KL}\left(\mathbf{q}_i^b \,\|\, \mathbf{s}_i^b\right)$

        `// Discrepancy-aware commitment & refinement (DCR)`

14        **for** $b \in \{g, \ell\}$ **do**

15           $\mathcal{Z}_b(\mathbf{z}_i^t) \leftarrow \arg\min_{\mathbf{p}_m \in \mathbf{P}^b} \|\mathbf{z}_i^t - \mathbf{p}_m^b\|_2^2; \quad$ compute reliability $w_i^b$ from prototype relations $\mathbf{Q}_b$;

16        $\mathcal{L}_{\text{DCR}} \leftarrow \sum_i \sum_b w_i^b \left(\|\mathbf{z}_i^t - \text{sg}[\mathcal{Z}_b(\mathbf{z}_i^t)]\|_2^2 + \|\text{sg}[\mathbf{z}_i^t] - \mathcal{Z}_b(\mathbf{z}_i^t)\|_2^2\right)$

        `// Overall objective and updates (teacher frozen)`

17        $\mathcal{L} \leftarrow \mathcal{L}_{\text{PSD}} + \lambda\, \mathcal{L}_{\text{DCR}}$

18        Update $S_g$, $\{S_p^\ell\}_{p=1}^N$, $\mathbf{W}_r$, $\phi$, and $\{\mathbf{P}_b\}$ by gradient descent on $\mathcal{L}$

19 **Stage C: Zero-shot inference on unseen graph** $\mathcal{G} = (\mathcal{V}, \mathcal{E}, \mathbf{X})$ **(no retraining)**

20 Compute $\mathbf{Z}_T$, $\mathbf{h}^g$, $\mathbf{h}^\ell$, soft labels $\{\mathbf{q}^b\}$, predictions $\{\mathbf{s}^b\}$, and quantizers $\mathcal{Z}_b(\cdot)$ as above

21 **for** $v_i \in V$ **do**

22     $s_i \leftarrow \sum_{b\in\{g,\ell\}} \left[ \text{KL}\left(\mathbf{q}_i^b \,\|\, \mathbf{s}_i^b\right) + \lambda\left(\|\mathbf{h}_i^b - \mathcal{Z}_b(\mathbf{h}_i^b)\|_2^2 + \|\mathbf{z}_i^t - \mathcal{Z}_b(\mathbf{z}_i^t)\|_2^2\right) \right]$

23 **return** $s(\cdot)$

---

and z-loss incur $\mathcal{O}(nN)$. Therefore, the overall training complexity of ProMoS is $\mathcal{O}\big(nKd^2 + ndN + ndM + dM^2\big)$.

Table 5 summarizes the complexity comparison with representative generalist GAD methods. In practice, the number of edges is usually much larger than the number of nodes, i.e., $m \gg n \gg \{K, N, M, K'\}$. Hence, the edge- and node-related terms dominate the overall cost, while contributions from the number of activated students $K$ (typically fixed to $K = 2$), the number of students $N$, and the number of prototypes $M$ can be regarded as negligible constants. Consequently, the complexity of ProMoS remains comparable to existing generalist GAD methods, while providing enhanced scalability and flexibility through its mixture-of-students design.

Table 5: Comparison of the complexity of generalist GAD methods. Here, $n = |\mathcal{V}|$ is the number of nodes, $m = |\mathcal{E}|$ is the number of edges, $d$ is the feature dimension, $N$ is the number of students, $K$ is the number of activated students, $M$ is the number of prototypes, $n_q$ is the number of query nodes and $n_k$ is the number of context nodes in ARC, and $K'$ is size of each graph prompt in UNPrompt.

| Method | Complexity |
|---|---|
| ARC (Liu et al., 2024) | $\mathcal{O}\big(nd^2 + ed + n_q n_k d + d\log d\big)$ |
| UNPrompt (Niu et al., 2025) | $\mathcal{O}\big(nd^2 + ed + K'nd\big)$ |
| AnomalyGFM (Qiao et al., 2025a) | $\mathcal{O}\big(md^2 + n^2 d + nd + d^2\big)$ |
| **ProMoS (Ours)** | $\mathcal{O}\big(nKd^2 + ndN + ndM + dM^2\big)$ |

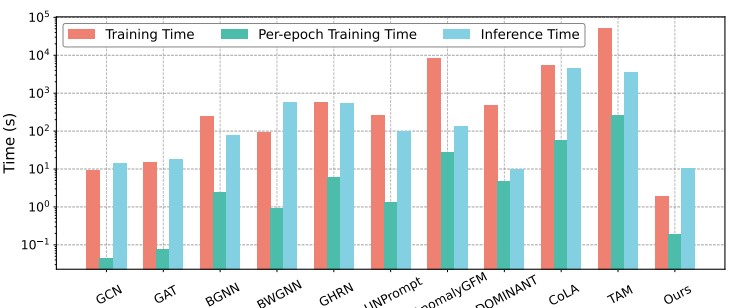

Figure 3: Time comparison with baseline.

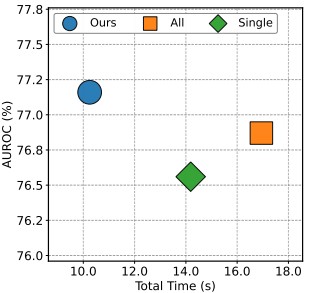

Figure 4: Performance comparison with MoS variants.

**Empirical Analysis.** As shown in Figure 3, in addition to reporting the overall training and inference time, we also provide the per-epoch training time and include comparisons with DOMINANT and TAM. Notably, DOMINANT runs out of memory on T-Finance, and thus its inference time excludes this dataset. Reconstruction-based methods, such as DOMINANT, require rebuilding the full adjacency matrix, making them infeasible for large graphs. TAM incurs even higher costs, as it relies on multi-round graph truncation and multi-network training, leading to substantial complexity in both training and inference. In contrast, ProMoS achieves remarkably low per-epoch training cost, second only to GCN and GAT, while delivering superior inference efficiency: it is 1.4× faster than GCN, the fastest baseline, and improves over existing GAD methods by orders of magnitude. These results further highlight ProMoS as an efficient and scalable solution for generalist GAD.

We further investigate the impact of different MoS architectures on both efficiency and performance. The *All* variant activates all student branches simultaneously, while the *Single* variant replaces the personalized branch with a single student of equivalent parameter count (achieved by widening its hidden layers). As shown in Figure 4, the sparsely-activated MoS achieves the best AUROC while maintaining the lowest training and inference cost. This demonstrates that dynamic sparsification enables more efficient parameter utilization, striking a favorable balance between expressiveness and efficiency.

# D  MORE EXPERIMENTAL SETUP

## D.1  DATASETS DETAILS

We evaluate on 15 benchmark datasets spanning diverse domains, as summarized in Table 6. To ensure broad coverage, the datasets are grouped into 8 categories: citation networks with injected anomalies (Cora, CiteSeer, ACM, PubMed), social networks with injected anomalies (BlogCatalog, Flickr), social networks with real anomalies (Facebook, Weibo, Reddit, Questions), co-review networks with real anomalies (YelpChi), co-author with injected anomalies (CoAuthor CS) and co-purchase networks with injected anomalies (Amazon Photo), crowd-sourcing service network with real anomalies (Tolokers) and finance networks with real anomalies (T-Finance).

Follow established protocols (Dong et al., 2024; Liu et al., 2024), in the first four categories that contain two datasets each, we select the larger graph (PubMed, Flickr, Questions, and YelpChi) as the training dataset. The remaining 11 graphs, including the smaller counterparts (Cora, CiteSeer, ACM, BlogCatalog, Facebook, Weibo, Reddit) and the four single-domain datasets (CoAuthor CS, Amazon Photo, Tolokers, T-Finance), are used for evaluation. This design enables us to examine both in-domain generalization, where the model is tested on datasets from the same domains as the training graph, and out-of-domain generalization, where it is tested on datasets from other domains. This diversity provides a comprehensive testbed to evaluate the robustness and adaptability of to unseen graphs. Specifically, the detailed descriptions for the datasets are given as follows:

- **Cora**, **CiteSeer**, **ACM** (Tang et al., 2008) and **PubMed** (Sen et al., 2008) are four citation networks, where nodes denote scientific publications and edges capture citation relationships between them. Each publication is described by a bag-of-words feature vector, with the vocabulary size determining the attribute dimension.

- **BlogCatalog** and **Flickr** (Ding et al., 2019; Tang & Liu, 2009) are representative social blog directories, where nodes correspond to users with inter-node links symbolizing mutual following. The node features are constructed from personalized textual content generated by users, such as blog posts or tagged images, reflecting individual interests and activities.

- **YelpChi** (Rayana & Akoglu, 2015; McAuley & Leskovec, 2013) is dataset about the relationship between users and reviews. YelpChi aims to identify anomalous reviews on Yelp.com that unfairly promote or demote products or businesses. Based on (Mukherjee et al., 2013; Rayana & Akoglu, 2015), three different graph datasets were derived from Yelp using different connections in user, product review text, and time. In this study, we focus on YelpChi-RUR (reviews posted by the same user).

- **Facebook** (Xu et al., 2022b), **Weibo** (Kumar et al., 2019), **Reddit** (Kumar et al., 2019) and **Questions** (Platonov et al., 2023) are four social networks with real anomalies. Facebook is a social network in which users can build relationships with others and share with their friends. Weibo dataset encompasses a graph of users and their associated hashtags from the Tencent Weibo platform. Suspicious behavior is defined by users posting multiple consecutive posts within a short temporal window (e.g., 60 seconds), with those exhibiting at least five such bursts labeled as anomalies. Node features combine geolocation information of the microblog post with bag-of-words features. Reddit serves as a forum posts network sourced from the social media platform Reddit, where users labeled as banned are identified as anomalies. Textual content from posts is encoded as vectors to serve as node attributes. Questions (Platonov et al., 2023) is constructed from Yandex Q, an online question-answering platform. Nodes correspond to users, and edges indicate whether a question–answer interaction occurred within a one-year period. Each user is represented by the averaged FastText embedding of their profile description, augmented with a binary feature flagging missing descriptions.

- **CoAuthor CS** (Shchur et al., 2018) are co-authorship graphs derived from the Microsoft Academic Graph (KDD Cup 2016). Nodes represent authors, and edges connect pairs of authors who have co-authored a paper. Node features represent paper keywords for each author's papers, and class labels indicate the most active fields of study for each author.

- **Amazon Photo** (Shchur et al., 2018) is a subgraph of the Amazon co-purchase network (McAuley et al., 2015), where nodes correspond to products and edges indicate frequent co-purchases. Each product is described by bag-of-words features extracted from reviews, with labels derived from product categories.

- **Tolokers** (Likhobaba et al., 2023) is constructed from the Toloka crowdsourcing platform. Nodes correspond to workers who participated in at least one of 13 selected projects, and edges link pairs of workers that completed the same task. Each node is described by profile attributes and task performance statistics, while labels indicate whether a worker was banned for anomalous behavior.

- **T-Finance** (Tang et al., 2022) is a large-scale transaction network where nodes represent anonymized accounts characterized by ten features capturing registration days, logging activities and interaction frequency. Edges connect pairs of accounts with transaction records. Human experts annotate nodes as anomalies if they fall into categories like fraud, money laundering and online gambling.

Table 6: The statistics of datasets.

| Dataset | Train | Test | #Nodes | #Edges | #Features | Avg. Degree | #Anomaly | %Anomaly |
|---|---|---|---|---|---|---|---|---|
| Citation network with injected anomalies | | | | | | | | |
| Cora | - | ✓ | 2,708 | 5,429 | 1,433 | 3.90 | 150 | 5.53 |
| CiteSeer | - | ✓ | 3,327 | 4,732 | 3,703 | 2.77 | 150 | 4.50 |
| ACM | - | ✓ | 16,484 | 71,980 | 8,337 | 8.73 | 597 | 3.62 |
| PubMed | ✓ | - | 19,717 | 44,338 | 500 | 4.50 | 600 | 3.04 |
| Social network with injected anomalies | | | | | | | | |
| BlogCatalog | - | ✓ | 5,196 | 171,743 | 8,189 | 66.11 | 300 | 5.77 |
| Flickr | ✓ | - | 7,575 | 239,738 | 12,047 | 63.30 | 450 | 5.94 |
| Social network with real anomalies | | | | | | | | |
| Facebook | - | ✓ | 1,081 | 55,104 | 576 | 50.97 | 25 | 2.31 |
| Weibo | - | ✓ | 8,405 | 407,963 | 400 | 48.53 | 868 | 10.30 |
| Reddit | - | ✓ | 10,984 | 168,016 | 64 | 15.30 | 366 | 3.33 |
| Questions | ✓ | - | 48,921 | 153,540 | 301 | 3.13 | 1,460 | 2.98 |
| Co-review network with real anomalies | | | | | | | | |
| YelpChi | ✓ | - | 23,831 | 49,315 | 32 | 2.07 | 1,217 | 5.10 |
| Co-author network with injected anomalies | | | | | | | | |
| CoAuthor CS | - | ✓ | 18,333 | 163,788 | 6,805 | 8.93 | 600 | 3.27 |
| Co-purchase network with injected anomalies | | | | | | | | |
| Amazon Photo | - | ✓ | 7,650 | 238,162 | 745 | 31.13 | 450 | 5.88 |
| Crowd-sourcing Service network with real anomalies | | | | | | | | |
| Tolokers | - | ✓ | 11,758 | 519,000 | 10 | 44.14 | 2,566 | 21.82 |
| Finance network with real anomalies | | | | | | | | |
| T-Finance | - | ✓ | 39,357 | 21,222,543 | 10 | 539.23 | 1,803 | 4.58 |

**Anomaly Injection.** To evaluate the effectiveness of our method in detecting diverse types of anomalies, we follow established practices (Song et al., 2007; Ding et al., 2019; Liu et al., 2021) by injecting an equal number of attributive and structural anomalies into each of the six datasets (ACM, PubMed, BlogCatalog, Flickr, CoAuthor CS, and Amazon Photo). Specifically, for structural anomalies, in a small clique, a small set of nodes are much more closely linked to each other than average, aligning with typical structural abnormalities observed in real-world networks (Skillicorn, 2007). To simulate such anomalies, we commence by defining the clique size $p$ and the number of cliques $q$. When generating a clique, $p$ nodes are randomly selected from the node set $\mathcal{V}$ and fully connected, thus marking all selected nodes $p$ as structural anomaly nodes. This process is iterated $q$ times to generate $q$ cliques, resulting in a total injection of $p \times q$ structural anomalies. In particular, we fix $p = 15$ and $q = 5, 5, 20, 20, 10, 15, 20, 15$ on Cora, CiteSeer, ACM, PubMed, BlogCatalog, Flickr, CoAuthor CS, and Amazon Photo, respectively. For feature anomalies, inconsistencies between node features and their neighboring contexts represent another prevalent anomaly observed in real-world scenarios. Following the pattern introduced by (Song et al., 2007), feature anomalies are created by perturbing node attributes. When generating a single feature anomaly, we first select a target node $v_i$ and then sample a set of $k$ nodes as candidates. Subsequently, from the candidate set, we choose the node $v_j$ with the maximum Euclidean distance from the feature of the target node $v_i$, and replace $v_i$'s feature with that of $v_j$. Here, we set $k = 50$ to ensure a sufficiently large perturbation magnitude. To ensure an equal balance in the quantity of both anomaly types, we set the number of feature anomalies to $p \times q$, implying that the above operation is repeated $p \times q$ times to generate all feature anomalies.

## D.2 IMPLEMENTATION DETAILS

**Metrics.** Following prior GAD protocols (Ding et al., 2019; Liu et al., 2021; 2024; Niu et al., 2025; Qiao et al., 2025a), we report AUROC and AUPRC, two standard metrics for anomaly detection. AUROC is obtained by ranking nodes according to their anomaly scores and computing the area under the ROC curve, while AUPRC is measured as the area under the precision-recall curve. Higher AUROC and AUPRC values indicate better detection performance. All results are averaged over five independent runs with different random seeds, and reported as mean±std.

**Computing infrastructure.** Experiments are carried out on a workstation running Ubuntu 22.04. The machine is equipped with an AMD EPYC 7542 processor (32 cores), a NVIDIA RTX 4090 GPU with CUDA 12.2 support, and 500 GiB of system memory.

**Software stack.** We used Python 3.9.22, PyTorch 2.3.1+cu121, torchvision 0.18.1+cu121, torchaudio 2.3.1+cu121, and DGL 0.9.0. For PyG we used torch-geometric 2.6.1 with torch-scatter 2.1.2+pt23cu121, torch-sparse 0.6.18+pt23cu121, torch-cluster 1.6.3+pt23cu121, and torch-spline-conv 1.2.2+pt23cu121.

**Implementation details.** We adopt four representative pre-trained graph SSL teacher models, namely GCA (Zhu et al., 2021) (default), GraphCL (You et al., 2020), BGRL (Thakoor et al., 2022), and DGI (Veličković et al., 2019), using the official implementations and hyperparameter settings provided by the PyG-SSL toolkit (Zheng et al., 2024). In addition, to prevent data leakage, the four representative graph SSL models are pre-trained exclusively on the training graphs $\mathcal{T}_{\text{train}}$. For each teacher model pre-training across multiple graphs, we follow established protocols (Dong et al., 2024; Liu et al., 2024): in each pre-training epoch, the teacher model is trained *sequentially* on the four graphs in $\mathcal{T}_{\text{train}}$. This ensures that the teacher model is trained across all training graphs following standard multi-graph SSL procedures. All baselines are configured to follow the same multi-graph pre-training paradigm for a fair comparison. For prototype initialization, we collect the node features from all nodes in the training graphs $\mathcal{T}_{\text{train}}$, concatenate them into a single feature matrix, and apply FAISS k-means to derive the initial prototypes used by ProMoS. For the rest of ProMoS, we conduct small grid sweeps around the default for all hyperparameters. The learning rate spans $\{10^{-2}, 5 \times 10^{-2}, 10^{-3}, 2 \times 10^{-3}, 5 \times 10^{-3}, 10^{-4}, 5 \times 10^{-4}, 10^{-5}\}$; the number of training epochs is $\{5, 10, 25, 50, 75, 100, 150, 200\}$; the trade-off parameter $\lambda$ is $\{0.1, 0.3, 0.5, 0.7, 1.0, 1.2, 1.5, 2.0\}$; the number of students $N$ is $\{2, 5, 10, 15, 20, 25, 30, 50\}$; the number of prototypes $M_b$ is $\{2, 5, 10, 15, 20, 25, 30, 50\}$; the routed activate top-$K$ students is $\{2, 4, 6, 8, 10, 12, 16, 20\}$; the sharpness $\beta$ is $\{0.2, 0.4, 0.6, 0.8, 1.0, 1.2, 1.6, 2.0\}$; the margin $\mu$ is $\{0.1, 0.2, 0.4, 0.5, 0.6, 0.8, 0.9, 1.0\}$; and the temperature $\tau$ is $\{0.2, 0.5, 0.7, 1.0, 1.5, 2.0, 2.5, 3.0\}$. After tuning, we adopt the configuration with learning rate $5 \times 10^{-3}$, epochs is 10, trade-off parameter $\lambda = 0.5$, number of students $N$ is 20, number of prototypes $M_B$ is 20, top-$K$ is 2, sharpness $\beta = 1$, margin $\mu = 0.6$, temperature $\tau = 2$.

# E SUPPLEMENTAL EXPERIMENTS

## E.1 GENERALIST GAD PERFORMANCE W.R.T. AUPRC

To mitigate potential bias from relying on a single evaluation metric, we further report results in terms of AUPRC across 11 datasets, comparing ProMoS with both supervised and unsupervised baselines. The experimental results are shown in Table 7, and the observations are similar to Table 2. First, ProMoS achieves the best performance on 9 out of 11 datasets and ranks second on the remaining 2 datasets, demonstrating its ability to more faithfully capture transferable normal patterns for anomaly detection. Second, on several datasets, unsupervised methods even outperform supervised ones, underscoring the inherent difficulty of anomaly detection. Since anomalies manifest differently across datasets, training classifiers to fit anomaly decision boundaries may compromise generalization to unseen graphs. By contrast, normal patterns are more stable and transferable across graphs. These results highlight the promise of unsupervised approaches that exploit unlabeled data to uncover universal normality as a foundation for generalizable GAD.

## E.2 COMPARISON WITH FEW-SHOT GENERALIST GAD

We further compare our method with two representative few-shot generalist methods, ARC and AnomalyGFM, under the 10-shot setting. Both ARC and AnomalyGFM rely on supervised pre-training on the training graphs and require access to 10 labeled nodes from the unseen test graph at inference, whereas ProMoS remains fully unsupervised throughout. As shown in Table 8, AnomalyGFM benefits noticeably from few-shot inference compared to its zero-shot performance. Nevertheless, ProMoS achieves the best AUROC on 6 out of 11 datasets and ranks second on the remaining 5, yielding a higher overall average than all few-shot baselines. These results underscore that our method not only delivers state-of-the-art performance, but also eliminates the reliance on scarce and costly labels, thereby greatly simplifying practical deployment.

Table 7: Anomaly detection performance on eleven datasets under the zero-shot setting. We report the mean and standard deviation of AUPRC. **1st** marks the best result, **2nd** the runner-up, and **3rd**."OOM" denotes out-of-memory.

| Method | Cora | CiteSeer | ACM | BlogCatalog | Facebook | Weibo |
|---|---|---|---|---|---|---|
| Supervised - Pre-Train Only | | | | | | |
| GCN | $7.41_{\pm 1.55}$ | $6.40_{\pm 1.40}$ | $5.27_{\pm 1.12}$ | $7.44_{\pm 1.07}$ | $1.59_{\pm 0.11}$ | $67.21_{\pm 15.20}$ |
| GAT | $6.49_{\pm 0.84}$ | $5.58_{\pm 0.62}$ | $4.70_{\pm 0.75}$ | $12.81_{\pm 2.08}$ | $3.14_{\pm 0.37}$ | $33.34_{\pm 9.80}$ |
| BGNN | $4.90_{\pm 1.27}$ | $3.91_{\pm 1.01}$ | $3.48_{\pm 1.33}$ | $5.73_{\pm 1.47}$ | $3.81_{\pm 2.12}$ | $30.26_{\pm 29.98}$ |
| BWGNN | $7.25_{\pm 0.80}$ | $6.35_{\pm 0.73}$ | $7.14_{\pm 0.20}$ | $8.99_{\pm 1.12}$ | $2.54_{\pm 0.63}$ | $12.13_{\pm 0.71}$ |
| GHRN | $9.56_{\pm 2.40}$ | $7.79_{\pm 2.01}$ | $5.61_{\pm 0.71}$ | $10.94_{\pm 2.56}$ | $2.41_{\pm 0.62}$ | $28.53_{\pm 7.38}$ |
| UNPrompt | $5.94_{\pm 0.77}$ | $5.00_{\pm 0.85}$ | $9.05_{\pm 1.32}$ | $23.93_{\pm 4.47}$ | $3.17_{\pm 0.44}$ | $16.04_{\pm 5.91}$ |
| AnomalyGFM | $5.11_{\pm 0.07}$ | $4.09_{\pm 0.11}$ | $4.01_{\pm 0.22}$ | $5.57_{\pm 0.30}$ | $6.02_{\pm 0.10}$ | $6.79_{\pm 0.04}$ |
| Unsupervised - Pre-Train Only | | | | | | |
| DOMINANT | $12.75_{\pm 0.71}$ | $13.85_{\pm 2.34}$ | $15.59_{\pm 2.69}$ | $35.22_{\pm 0.87}$ | $2.95_{\pm 0.06}$ | $81.47_{\pm 0.22}$ |
| CoLA | $11.41_{\pm 3.51}$ | $8.33_{\pm 3.73}$ | $7.31_{\pm 1.45}$ | $6.04_{\pm 0.56}$ | $1.90_{\pm 0.68}$ | $7.59_{\pm 3.26}$ |
| HCM-A | $5.78_{\pm 0.76}$ | $4.18_{\pm 0.75}$ | $4.01_{\pm 0.61}$ | $6.89_{\pm 0.34}$ | $2.08_{\pm 0.60}$ | $21.91_{\pm 11.78}$ |
| TAM | $11.18_{\pm 0.75}$ | $11.55_{\pm 0.44}$ | $23.20_{\pm 2.36}$ | $10.57_{\pm 1.17}$ | $8.40_{\pm 0.97}$ | $16.46_{\pm 0.09}$ |
| Unsupervised - Pre-Train Only | | | | | | |
| Ours | $46.48_{\pm 0.30}$ | $46.91_{\pm 0.30}$ | $41.47_{\pm 0.20}$ | $36.20_{\pm 0.10}$ | $6.42_{\pm 0.34}$ | $72.05_{\pm 0.11}$ |

| Method | Reddit | CS | Photo | Tolokers | T-Finance | Avg. |
|---|---|---|---|---|---|---|
| Supervised - Pre-Train Only | | | | | | |
| GCN | $3.39_{\pm 0.39}$ | $2.78_{\pm 0.02}$ | $5.27_{\pm 0.08}$ | $20.81_{\pm 0.16}$ | $9.82_{\pm 0.18}$ | $12.49_{\pm 18.87}$ |
| GAT | $3.73_{\pm 0.54}$ | $2.97_{\pm 0.09}$ | $5.62_{\pm 0.17}$ | $22.87_{\pm 0.98}$ | $9.93_{\pm 0.11}$ | $10.11_{\pm 9.66}$ |
| BGNN | $3.52_{\pm 0.50}$ | $4.19_{\pm 0.89}$ | $5.96_{\pm 0.18}$ | $24.46_{\pm 2.43}$ | $4.57_{\pm 0.59}$ | $8.62_{\pm 9.39}$ |
| BWGNN | $3.69_{\pm 0.81}$ | $8.09_{\pm 0.62}$ | $6.55_{\pm 0.60}$ | $22.60_{\pm 1.41}$ | $5.02_{\pm 0.64}$ | $8.21_{\pm 5.42}$ |
| GHRN | $3.24_{\pm 0.33}$ | $5.79_{\pm 1.16}$ | $7.67_{\pm 1.91}$ | $23.79_{\pm 5.74}$ | $8.13_{\pm 7.57}$ | $10.31_{\pm 8.29}$ |
| UNPrompt | $4.12_{\pm 0.22}$ | $9.88_{\pm 1.02}$ | $5.99_{\pm 0.57}$ | $16.86_{\pm 1.11}$ | $2.79_{\pm 0.27}$ | $9.34_{\pm 6.82}$ |
| AnomalyGFM | $3.99_{\pm 0.11}$ | $2.84_{\pm 0.06}$ | $5.32_{\pm 0.12}$ | $21.39_{\pm 0.75}$ | $7.65_{\pm 2.31}$ | $6.51_{\pm 5.35}$ |
| Unsupervised - Pre-Train Only | | | | | | |
| DOMINANT | $3.49_{\pm 0.44}$ | $15.48_{\pm 0.46}$ | $7.88_{\pm 0.04}$ | $20.51_{\pm 0.25}$ | *OOM* | $20.92_{\pm 23.20}$ |
| CoLA | $3.71_{\pm 0.67}$ | $3.66_{\pm 0.14}$ | $6.40_{\pm 0.45}$ | $22.93_{\pm 1.47}$ | $4.61_{\pm 0.36}$ | $7.63_{\pm 5.71}$ |
| HCM-A | $3.18_{\pm 0.23}$ | $26.56_{\pm 12.10}$ | $6.73_{\pm 0.61}$ | $20.56_{\pm 1.15}$ | *OOM* | $10.19_{\pm 9.09}$ |
| TAM | $3.94_{\pm 0.13}$ | $29.51_{\pm 1.46}$ | $15.46_{\pm 1.29}$ | $23.32_{\pm 0.08}$ | $5.91_{\pm 0.67}$ | $14.50_{\pm 8.01}$ |
| Unsupervised - Pre-Train Only | | | | | | |
| Ours | $4.85_{\pm 0.02}$ | $33.04_{\pm 0.07}$ | $18.33_{\pm 0.83}$ | $27.37_{\pm 0.63}$ | $10.16_{\pm 0.96}$ | $31.21_{\pm 20.53}$ |

### E.3 ADAPTABILITY OF PROMOS TO DIFFERENT TEACHERS

We evaluate the adaptability of ProMoS to different teacher models by comparing several well-established graph SSL methods across different paradigms. We first examine four representative contrastive teachers: GCA (Zhu et al., 2021) (default), GraphCL (You et al., 2020), BGRL (Thakoor et al., 2022), and DGI (Veličković et al., 2019), using the official implementations and recommended hyperparameters from the PyG-SSL Toolkit (Zheng et al., 2024). We then include the reconstruction-based GraphMAE (Hou et al., 2022), implemented using its released source code. As reported in Table 4, all variants of ProMoS achieve strong and stable performance across the six datasets. ProMoS+GCA, ProMoS+GraphCL, and ProMoS+BGRL each achieve the best performance on two datasets, while ProMoS+DGI consistently delivers the second-best results overall. The reconstruction-based ProMoS+GraphMAE also achieves competitive performance, demonstrating that ProMoS remains effective even when paired with a teacher from a different learning paradigm. These findings demonstrate the robustness of our framework to the choice of teacher and highlight its plug-and-play nature, allowing future substitution with more advanced graph SSL models as they emerge.

Table 8: AUROC (%, mean±std over five runs) on eleven datasets, comparing with supervised pre-training methods using few-shot inference (ARC (Liu et al., 2024) and AnomalyGFM (Qiao et al., 2025a)), while ours is fully unsupervised and pre-trained only. **1st** marks the best result, **2nd** the runner-up, and **3rd**.

| Method | Cora | CiteSeer | ACM | BlogCatalog | Facebook | Weibo |
|---|---|---|---|---|---|---|
| Supervised - Pre-Train & Few-Shot Inference | | | | | | |
| ARC | $87.45_{\pm 0.74}$ | $90.95_{\pm 0.59}$ | $79.88_{\pm 0.28}$ | $74.76_{\pm 0.06}$ | $67.56_{\pm 1.60}$ | $88.85_{\pm 0.14}$ |
| AnomalyGFM | $56.31_{\pm 1.11}$ | $51.27_{\pm 1.48}$ | $64.03_{\pm 0.93}$ | $67.68_{\pm 0.99}$ | $75.66_{\pm 2.46}$ | $56.91_{\pm 2.55}$ |
| Unsupervised - Pre-Train Only | | | | | | |
| Ours | $84.56_{\pm 0.16}$ | $90.77_{\pm 0.12}$ | $89.47_{\pm 0.79}$ | $76.17_{\pm 0.37}$ | $69.31_{\pm 0.50}$ | $91.74_{\pm 0.03}$ |

| Method | Reddit | CS | Photo | Tolokers | T-Finance | Avg. |
|---|---|---|---|---|---|---|
| Supervised - Pre-Train & Few-Shot Inference | | | | | | |
| ARC | $60.04_{\pm 0.69}$ | $82.73_{\pm 0.48}$ | $75.55_{\pm 0.72}$ | $53.42_{\pm 2.70}$ | $64.10_{\pm 3.10}$ | $75.03_{\pm 12.45}$ |
| AnomalyGFM | $50.18_{\pm 3.52}$ | $50.99_{\pm 1.15}$ | $61.19_{\pm 0.84}$ | $59.36_{\pm 5.44}$ | $56.97_{\pm 4.04}$ | $59.14_{\pm 7.75}$ |
| Unsupervised - Pre-Train Only | | | | | | |
| Ours | $60.83_{\pm 0.35}$ | $88.85_{\pm 0.60}$ | $72.67_{\pm 1.07}$ | $52.80_{\pm 0.82}$ | $71.62_{\pm 1.06}$ | $77.16_{\pm 13.09}$ |

Table 9: Comparison of optimization strategies for the commitment and refinement loss.

| Method | ACM | Facebook | Reddit | CS | Photo | T-Finance | Avg. |
|---|---|---|---|---|---|---|---|
| ProMoS | $89.47_{\pm 0.79}$ | $69.31_{\pm 0.50}$ | $60.83_{\pm 0.35}$ | $88.85_{\pm 0.60}$ | $72.67_{\pm 1.07}$ | $71.62_{\pm 1.06}$ | **75.46** |
| Alt-PT | $89.67_{\pm 1.00}$ | $69.19_{\pm 0.57}$ | $60.73_{\pm 0.66}$ | $88.95_{\pm 1.08}$ | $72.80_{\pm 1.36}$ | $70.58_{\pm 0.65}$ | 75.32 |
| Alt-TP | $89.86_{\pm 0.65}$ | $68.95_{\pm 0.50}$ | $60.62_{\pm 0.98}$ | $89.16_{\pm 0.82}$ | $71.19_{\pm 1.69}$ | $70.39_{\pm 1.15}$ | 75.03 |

### E.4 COMPARISON OF OPTIMIZATION STRATEGIES FOR THE COMMITMENT AND REFINEMENT LOSS

The commitment and refinement module serves two purposes: it regularizes the teacher's feature space toward the prototype space and simultaneously refines the prototypes to capture transferable, high-level semantics. In this section, we evaluate the influence of optimizing these two components simultaneously on prototype distinctiveness by overly coupling their learning dynamics. To examine this, we compare the default *joint* optimization strategy in ProMoS with two epoch-level *alternating* strategies: (i) **Alt-PT**, which first updates the prototypes and then applies the commitment loss to regularize the teacher features; and (ii) **Alt-TP**, which first regularizes the teacher features and then updates the prototypes.

Table 9 summarizes the results across six datasets. All three strategies yield highly consistent performance, and the joint optimization used in ProMoS achieves the best average AUC. These findings indicate that simultaneous optimization does not compromise prototype distinctiveness; instead, the joint strategy remains a stable and effective choice for aligning the teacher feature space with the evolving prototype semantics.

### E.5 COMPARISON OF PRETRAINING DATASET CONFIGURATIONS

To ensure fairness, our main experiments follow the ARC protocol and pre-train the teacher models on PubMed, Flickr, Questions, and YelpChi—the configuration adopted in the first generalist GAD study. To assess the influence of the choice of pretraining datasets on downstream performance, we further evaluate an alternative configuration (Alt-DPD) in which the teacher is pre-trained on a different set of graphs: Cora, CiteSeer, ACM, and PubMed.

Table 10 reports the results across six target graphs. The alternative configuration achieves performance highly comparable to that of ProMoS, with average AUCs of 72.70 and 72.50, respectively. These results indicate that ProMoS is not sensitive to the specific selection of pretraining datasets and generalizes well across different pretraining regimes.

Table 10: Comparison of pretraining dataset configurations.

| Method | Facebook | Weibo | Reddit | CS | Photo | Tolokers | Avg. |
|--------|----------|-------|--------|-----|-------|----------|------|
| ProMoS | $69.31_{\pm 0.50}$ | $91.74_{\pm 0.03}$ | $60.83_{\pm 0.35}$ | $88.85_{\pm 0.60}$ | $72.67_{\pm 1.07}$ | $52.80_{\pm 0.82}$ | **72.70** |
| Alt-DPD | $68.58_{\pm 0.38}$ | $91.77_{\pm 0.07}$ | $58.15_{\pm 1.58}$ | $89.15_{\pm 0.32}$ | $73.39_{\pm 1.15}$ | $53.96_{\pm 0.66}$ | 72.50 |

### E.6 VISUALIZATION ANALYSIS

To further examine the effectiveness of the proposed components, we provide three additional visualization studies in Figure 5.

**Effectiveness of the commitment and refinement losses.** Figure 5a and Figure 5b visualize the standardized teacher outputs on an unseen Cora graph using t-SNE. Normal nodes (blue circles) and anomalous nodes (red squares) are plotted together with the shared and personalized prototypes projected into the same embedding space. Comparing the model without the discrepancy-aware commitment and refinement loss (w/o $\mathcal{L}_{DCR}$) to the model with it (w/ $\mathcal{L}_{DCR}$) reveals a clear contrast: w/o $\mathcal{L}_{DCR}$ (Figure 5a), the teacher features become scattered and unstructured under distribution shift, and the prototypes lose semantic anchoring. In contrast, w/ $\mathcal{L}_{DCR}$ (Figure 5b), both shared and personalized prototypes remain discriminative and representative, and anomalous nodes are more likely to be pushed away from any prototype regions. These observations indicate that the proposed DCR mechanism optimizes the prototype space and regularizes the teacher representation space, thereby strengthening out-of-distribution robustness.

**Effectiveness of the Two-Branch MoS Architecture.** Figure 5b further illustrates the distinct yet complementary roles of the shared and personalized branches in our Mixture-of-Students architecture. The shared prototypes (triangles) consistently lie near the central, high-density semantic regions, capturing global normality patterns that are transferable across graphs. In contrast, the personalized prototypes (inverted triangles) capture local patterns, often spreading to peripheral or fine-grained regions that the shared branch cannot model. This division of labor aligns with the intended design of MoS: the shared branch provides a universal backbone of normality, while personalized students specialize in diverse patterns. Moreover, the ablation results in Table 3 reinforce this observation. Eliminating either branch leads to performance degradation, demonstrating that both components are indispensable for capturing the full spectrum of normality patterns required for robust generalist GAD.

**Teacher–Student Feature Distributions** Figure 5c illustrates the projections of the teacher (blue circles) and student (yellow triangles) representations on the unseen CiteSeer graph. Overall, the two distributions exhibit no substantial global gap, indicating that the student network is able to approximate the teacher's representation space even under distribution shift. To obtain a more fine-grained view, we randomly sample 20 normal teacher–student pairs and 20 anomalous teacher–student pairs, where each pair consists of a node's teacher embedding and the corresponding student embedding, and connect each pair with a line segment.

The visualization reveals a clear pattern. Normal pairs (green lines) show consistently shorter teacher–student distances: their embeddings align closely in the representation space, demonstrating that the MoS architecture can better reconstruct the teacher's normality patterns. In contrast, anomalous pairs (red lines) exhibit significantly larger distortions. Due to their irregular semantics and weaker prototype affinity, anomalous nodes cannot be faithfully reconstructed by the students, resulting in pronounced divergence in their representations. This behavior aligns precisely with our anomaly scoring mechanism, where reconstruction deviation serves as a crucial indicator of abnormality. These results confirm that the student network closely matches the teacher distribution for normal nodes while naturally amplifying deviations on anomalous nodes—exactly the behavior desired for zero-shot anomaly detection.

**Visual Evidence of Student Diversity** To directly examine whether personalized students capture different modes, we visualize student-specific embeddings on the Weibo dataset (Figure 5d). For each of the 20 students, we extract the personalized-branch outputs and project them into a two-dimensional space, with points colored by student index. The resulting clusters are clearly separated across students, indicating that each student focuses on distinct semantic patterns in the feature

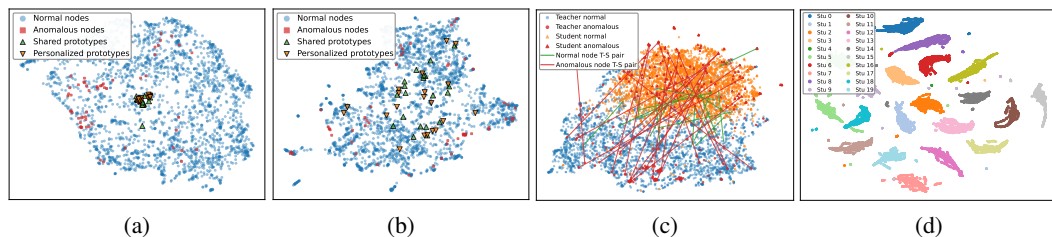

(a)  (b)  (c)  (d)

Figure 5: Visualization analysis of ProMoS. The four subfigures respectively show: (a) the embedding distributions w/o $\mathcal{L}_{\text{DCR}}$ on Cora; (b) the embedding distributions w/ $\mathcal{L}_{\text{DCR}}$ on Cora; (c) the teacher–student feature alignment on CiteSeer; and (d) the student-specific embedding on Weibo.

space. This empirical pattern is consistent with the design of ProMoS. The diversity among student models is structurally encouraged by the sparse Top-K routing mechanism: instead of sending every sample to every student, Top-K routing assigns each student only a subset of nodes, so semantically similar nodes are routed to partially overlapping students while each student is trained on a different portion of the graph. This routing scheme naturally guides different students toward different subsets of data and fosters meaningful specialization without requiring an explicit diversity regularizer.

### E.7  HYPERPARAMETER SENSITIVITY ANALYSIS

We evaluate the sensitivity of key hyperparameters on ProMoS and report the average AUROC and AUPRC on an unseen test graphs.

**Effect of trade-off parameter $\lambda$.** Trade-off parameter $\lambda$ balances prototype-guided soft-label distillation (PSD) and discrepancy-aware commitment & refinement (DCR). As shown in the figure 6a, small $\lambda$ leads to insufficient teacher constraints and prototype updates, while large $\lambda$ overconstrains the student and affects distillation performance. Therefore, we choose $\lambda = 0.5$ by default.

**Effect of number of students $N$.** In the mixture-of-students (MoS) architecture, the number of students $N$ determines how many personalized branches are initialized. As shown in Figure 6b, increasing $N$ initially improves both AUROC and AUPRC. With too few students, the model lacks sufficient expressive capacity to capture all normal patterns encoded in the teacher, resulting in underfitting. As $N$ continues to grow, the performance gains saturate and parameter efficiency diminishes, since additional students contribute marginally to representation diversity. In our experiments, we set the default number of students to $N = 20$.

**Effect of number of prototypes $M_b$.** As shown in Figure 6c, increasing $M_b$ from very small values to a mid-to-high range improves performance, after which the results become stable. Too few prototypes lead to underfitting of transferable semantics with overly coarse granularity, while too many prototypes fragment the space, slightly reducing robustness and increasing inference time. A mid-range $M_b$ offers the best trade-off.

**Effect of Top-$K$ activated students.** As shown in Figure 6d, activating fewer students (e.g., $K = 2, 4, 6$) achieves the best AUROC/AUPRC. Larger $K$ values dilute the distillation signals each student receives and prevent individual students from specializing in distinct patterns. This undermines the divide-and-conquer principle of MoS, causing different students to converge toward similar behaviors and ultimately degrading performance.

**Effect of Sharpness $\beta$ and margin $\mu$.** Both parameters regulate the discrepancy-aware weighting by controlling how strongly unreliable samples are down-weighted. As shown in Figure 6e and 6f, moderate values yield the best AUROC/AUPRC. With too small $\beta$, the weighting is overly soft, making reliable and unreliable samples indistinguishable. Increasing $\beta$ sharpens the reweighting and improves robustness, but excessive sharpness overemphasizes a few samples and amplifies noise, leading to performance drops. For $\mu$, a small margin down-weights many diverse yet reliable, informative nodes (with modest KL), while an excessively large $\mu$ inflates the weights of high-KL nodes—including unreliable or anomalous nodes—thereby degrading robustness. Hence, performance peaks at moderate $\beta$ and $\mu$.

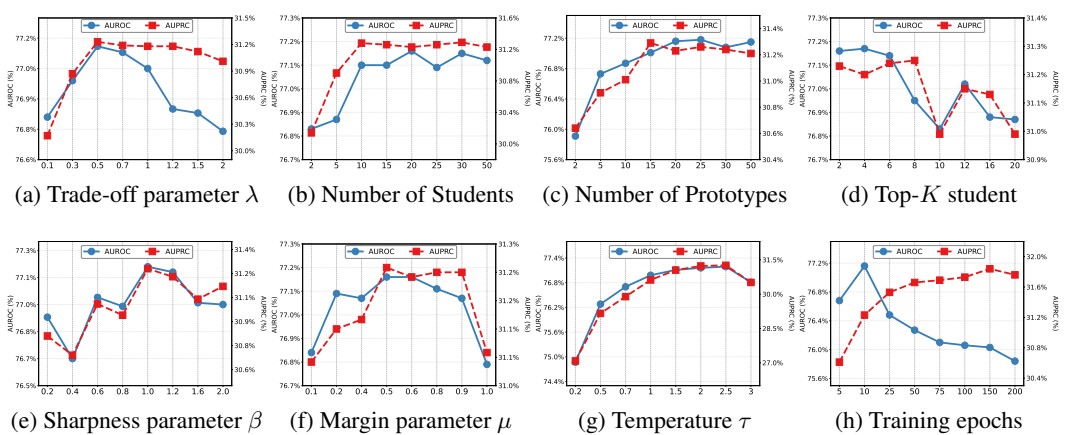

Figure 6: The hyperparameter sensitivity analysis of ProMoS.

**Effect of temperature $\tau$.** In Prototype-guided Soft-label Distillation (PSD), the temperature coefficient $\tau$ controls the smoothness of the teacher's soft labels. As shown in Figure 6g, a very low $\tau$ produces overly confident prototype posteriors, making the soft labels too sharp and limiting the transfer of richer semantic information. In contrast, a very high $\tau$ yields overly flat targets, which weakens the guidance signal and reduces the effectiveness of distillation. Moderate values of $\tau$ strike a balance, preserving informative relative probabilities while avoiding overconfidence, and thus achieve the best performance.

**Effect of training epochs.** Interestingly, we observe a clear divergence between AUROC and AUPRC (Figure 6h). AUROC reaches its peak in the early stage of training and then gradually declines, whereas AUPRC continues to improve until it flattens out. This behavior arises because our method primarily models normal patterns: with prolonged training, the decision boundary becomes increasingly fitted to normal nodes, effectively shrinking the boundary. Such refinement benefits AUPRC, which is more sensitive to anomaly nodes, but slightly reduces AUROC by compressing the global ranking margins—some normal nodes near the boundary may be assigned higher anomaly scores, lowering the overall ranking quality. In practice, the choice of training epochs should be guided by the downstream evaluation priority: if overall ranking is crucial, early stopping is preferable, whereas if precision and recall of anomalies are emphasized, longer training may be beneficial.

# F  USE OF LARGE LANGUAGE MODELS

LLMs are employed only to aid writing clarity and polish. Importantly, all core scientific contributions, including problem formulation, model design, theoretical analysis, and experiments, are entirely conceived and executed by the authors. The authors take full responsibility for all technical content, claims, and conclusions presented in this work.

