# OpenReview forum: "ProMoS: Prototype-Guided Distillation for Generalist Graph Anomaly Detection"
_ICLR.cc/2026/Conference — Submitted to ICLR 2026_

### Official Review · Reviewer_tmaC · 2025-10-25

**Soundness:** 3
**Presentation:** 2
**Contribution:** 3
**Rating:** 6
**Confidence:** 4

**Summary:**

This paper focuses on generalist graph anomaly detection and introduces a method named ProMoS. It is an unsupervised generalist GAD framework that consists of a self-supervised GNN teacher and a mixture-of-students. The model optimization is performed via prototype-guided soft-label distillation.

**Strengths:**

1. The paper is well-structured and the code is released.

2. The studied problem is practical and important.

3. Experiments are comprehensive, demonstrating the effectiveness of the proposed method.

**Weaknesses:**

1. The introduction of discrepancy-aware commitment and refinement is not very clear. More detailed descriptions are needed.

2. What does r_i mean in Eq.2?

3. Since UNPrompt and AnomalyGFM are pre-trained on one dataset originally, how are they pre-trained in the proposed setting?

4. The authors are encouraged to provide more analysis or visualizations to demonstrate the effectiveness of the two-branch design.

**Questions:**

Please see the weaknesses.

---

> ### Author Response · Authors · 2025-11-21
> **Author Response to Reviewer tmaC**
>
> Thank you for the positive feedback on the paper structure, practicality, and comprehensive experiments. Next we will reply to comments one by one.
>
> >W1: The introduction of discrepancy-aware commitment and refinement is not very clear. More detailed descriptions are needed.
>
> **R1:** Thank you for pointing this out. We have substantially revised and clarified the discrepancy-aware commitment and refinement component. Specifically, we:
>
> - **Clarify the objective’s role**: The commitment and refinement objective (Eq. 10) regularizes the teacher outputs to enforce a consistent and well-structured feature space across graphs, and simultaneously updates the prototypes to capture high-level, transferable semantics.
>
> - **Explain the reliability-aware weighting strategy**: Eq. 9 implements an adaptive weighting mechanism that assigns larger weights to high-quality samples and downweights unreliable ones. In the revised manuscript, we provide additional details and a clearer explanation of the intuition behind this design.
>
> - **Detail the loss design**: We define discrepancy-aware commitment and refinement losses, where the commitment loss pulls teacher features toward prototypes and the refinement loss updates prototypes to better reflect transferable semantic structure.
>
> We provide a clearer, step-by-step explanation of this mechanism in the revised subsection. In addition, we also provide complementary visualization analyses of the discrepancy-aware commitment and refinement losses in Appendix E.6, and more details on these visual results are available in our response to `Reviewer 6gsy (R3)`.
>
> >W2: What does r_i mean in Eq.2?
>
> **R2:** Thank you for highlighting this issue. In Eq. (2), $\mathbf{e}\_i$ was mistakenly written as $\mathbf{r}\_i$. The correct residual update is $\tilde{\mathbf{x}}\_i = \mathbf{x}\_i + \mathbf{e}\_i$. The symbol $\mathbf{r}\_i$ is used only later in Eq. (4) to denote the routing probability of node $i$. We have corrected this notation in the revised manuscript and carefully reviewed the entire method section to prevent similar inconsistencies.
>
> >W3: Since UNPrompt and AnomalyGFM are pre-trained on one dataset originally, how are they pre-trained in the proposed setting?
>
>
> **R3:** We would like to note that, to ensure fair comparison, we pre-train UNPrompt and AnomalyGFM using the same multi-graph protocol as ARC, i.e., iterating over all training graphs sequentially within each epoch rather than merging them into a single graph.
>
> - **For UNPrompt**, the multi-graph setting is directly supported by its official implementation [train.py](https://github.com/mala-lab/UNPrompt/blob/main/train.py), where the variable `traindatasets` is defined as a list of datasets. Therefore, no modification is required on our side: we simply populate traindatasets with all ARC training graphs.
> - **For AnomalyGFM**, which by default pre-trains on a single dataset, we lightly adjust its source code such that its training loop iterates over the full list of ARC training graphs in each epoch, mirroring the way UNPrompt handles multiple datasets.
>
>
> >W4: The authors are encouraged to provide more analysis or visualizations to demonstrate the effectiveness of the two-branch design.
>
> **R4:** In the revised manuscript, we add new analyses and visualizations in Appendix E.6 to directly assess the effectiveness of the two-branch design. These results show that the shared and personalized branches capture complementary normality patterns: the shared branch models global, transferable semantics, while the personalized branch focuses on more local or fine-grained variations. This division of labor is further supported by the ablation results in Table 3, where removing either branch leads to performance degradation, indicating that both components are essential for robust generalist GAD. Additional visualization of the commitment and refinement losses and teacher–student distribution differences is also provided in Appendix E.6.

---

> ### Author Response · Authors · 2025-11-27
>
> Dear Reviewer tmaC,
>
> We would like to express our sincere gratitude to you for reviewing our paper and providing valuable feedback. We believe that we have responded to and addressed all your concerns with our revisions — in light of this, we hope you consider raising your score.
>
> If you have any additional questions or would like further clarification on any point, we would be very glad to respond promptly. Thanks!
>
> Best,
>
> All authors

---

### Official Review · Reviewer_6V1N · 2025-10-29

**Soundness:** 3
**Presentation:** 2
**Contribution:** 3
**Rating:** 6
**Confidence:** 4

**Summary:**

The paper proposes ProMoS, an unsupervised generalist graph anomaly detection framework capable of zero-shot detection on unseen graphs. Specifically, it first builds a self-supervised GNN teacher and transfers normality representations to a mixture-of-students (MoS) model with local and global branches. Moreover, through prototype-guided soft-label distillation, ProMoS enhances cross-graph generalization.

**Strengths:**

1.	This paper focuses on unsupervised generalist graph anomaly detection, which is a challenging and practical problem. Moreover, the code is released.
2.	The utilization of knowledge distillation and prototypes alignment enhances cross-graph transferability and generalizability.
3.	The proposed method achieves better performance than the used baselines, demonstrating its effectiveness.

**Weaknesses:**

1.	For the pre-trained teacher, can it be replaced with other non-SSL methods?
2.	Does the shared branch and personalized branch share the prototypes as it says “share” in Figure 1? Moreover, how are they initialized, from teacher model or the student model?
3.	The authors should provide more analysis as to why Eq.9 could measure the reliability of nodes. For Eq.11, there is no routing regularization term.
4.	The authors use PubMed, Flickr, Questions and YelpChi as the pretraining datasets. Is there a specific reason? If not, what is the performance when using different pretraining datasets.
5.	In Table 5, the authors argue that it reports empirical runtimes. But only the complexity of methods is provided.

**Questions:**

Please see the weaknesses.

---

> ### Author Response · Authors · 2025-11-21
> **Author Response to Reviewer 6V1N (1/3)**
>
> We appreciate your positive feedback on the problem significance, method design, and experimental results. Next we will reply to comments one by one.
>
> >W1: For the pre-trained teacher, can it be replaced with other non-SSL methods?
>
>
> **R1:** The pre-trained teacher in our framework **must be unsupervised**, since the generalist GAD setting assumes no access to node-level labels during pretraining. Reconstruction-based methods offer another important unsupervised paradigm by reconstructing the graph structure or attributes, different from contrastive approaches. This type of method can certainly be considered as a teacher model. Here, we choose a representative method, GraphMAE [1], for experimental comparison. (If we have misinterpreted your question in any way, we would be grateful if you could clarify this in the next discussion.)
>
> As shown in the table below, ProMoS maintains comparable performance on most datasets, demonstrating its robustness to unsupervised pretraining paradigms and plug-and-play nature. However, the average performance is slightly lower than that obtained with contrastive SSL teachers (e.g., GCA), which aligns with the widely observed trend that reconstruction-based methods typically underperform contrastive approaches in graph representation learning [2]. Overall, while ProMoS can technically accommodate other paradigm teachers, we recommend using contrastive SSL methods in practice due to their stronger empirical performance and better transferability.
>
>
> | Method          | Cora                         | CiteSeer                     | ACM                          | BlogCatalog                  | Weibo                        | Tolokers                     | Avg.    |
> | --------------- | ---------------------------- | ---------------------------- | ---------------------------- | ---------------------------- | ---------------------------- | ---------------------------- | ------- |
> | ProMoS+GCA      | $84.56{\scriptstyle\pm0.16}$ | $90.77{\scriptstyle\pm0.12}$ | $89.47{\scriptstyle\pm0.79}$ | $76.17{\scriptstyle\pm0.37}$ | $91.74{\scriptstyle\pm0.03}$ | $52.80{\scriptstyle\pm0.82}$ | $80.92$ |
> | ProMoS+GraphMAE | $83.76{\scriptstyle\pm0.09}$ | $90.51{\scriptstyle\pm0.09}$ | $76.83{\scriptstyle\pm0.47}$ | $73.76{\scriptstyle\pm0.34}$ | $91.73{\scriptstyle\pm0.07}$ | $50.24{\scriptstyle\pm0.64}$ | $77.81$   |
>
> > W2: Does the shared branch and personalized branch share the prototypes as it says “share” in Figure 1? Moreover, how are they initialized, from teacher model or the student model?
>
> **R2:** The shared branch and the personalized branch each maintain their own prototype codebooks and **do not share** prototypes with one another. In Fig. 1, “share” indicates that the teacher and student models use the same shared and personalized prototype codebooks during distillation, not that the two branches share codebooks. Prototype initialization is performed by applying FAISS k-means to the concatenated node features from the four training graphs. Additional details on initialization and pretraining have been added to Appendix D.2.

---

> ### Author Response · Authors · 2025-11-21
> **Author Response to Reviewer 6V1N (2/3)**
>
> >W3: The authors should provide more analysis as to why Eq.9 could measure the reliability of nodes. For Eq.11, there is no routing regularization term.
>
> **R3:** Eq. 9 implements an unsupervised, quality-aware weighting mechanism that measures node reliability by checking how well a node’s teacher-induced prototype distribution aligns with the global semantic structure among prototypes. Specifically, we first construct the prototype–prototype relation matrix $\mathbf Q^{b} = \operatorname{softmax}\!\left(\mathrm{sim}(\mathbf P^{b}, \mathbf P^{b})/{\tau}\right)$, which provides a global semantic structure among prototypes and serves as the ground-truth relational pattern. For a node $i$, the row $\mathbf{Q}\_{m\_i^\star}^{b}$ corresponding to its nearest prototype pattern serves as the canonical reference. We compare the teacher-induced prototype distribution $\mathbf q\_i^{b}$ with this reference $\mathbf{Q}\_{m\_i^\star}^{b}$ using KL divergence, which measures their semantic consistency. High-quality normal nodes exhibit distributions that closely follow $\mathbf{Q}\_{m\_i^\star}^{b}$—yielding small KL divergence and thus large reliability weights—whereas noisy or anomalous nodes deviate from the global prototype structure and therefore receive reduced weights. We have substantially rewritten the discrepancy-aware commitment and refinement subsection to better highlight this intuition and to clarify the roles of the $\beta$ and $\mu$ parameters.
>
> Regarding Eq. 11, we initially experimented with incorporating an explicit routing regularization term. However, it consistently yielded negligible performance gains, so we removed it from the final objective for clarity. Moreover, as shown in the inference-stage student activation statistics reported in `Reviewer RTkz (R1)`, the expert usage of our MoS architecture is already well balanced in practice, suggesting that an additional routing regularizer offers limited benefit.
>
> For completeness, the routing regularization term we previously experimented with is defined as: $
> \mathcal{L}\_{\text{RR}} = N \cdot \sum\_{p=1}^{N} \bar{\mathbf{r}}\_{p} \cdot \bar{\mathbf{g}}\_{p},
> \quad
> \bar{\mathbf{r}}\_{p} = \tfrac{1}{|\mathcal{V}|} \sum\_{i \in \mathcal{V}} \mathbf{r}\_{i}[p],
> \quad
> \bar{\mathbf{g}}\_{p} = \tfrac{1}{|\mathcal{V}|} \sum\_{i \in \mathcal{V}} \mathbb{I}\!\left[p \in \mathrm{Top}\text{-}K(\mathbf{r}\_{i}[1:N])\right],
> $
> where $N$ is the number of students and $\mathbb{I}[\cdot]$ is an indicator function. $\bar{\mathbf{r}}\_{p}$ denotes the average routing probability of student $p$, while $\bar{\mathbf{g}}\_{p}$ corresponds to its actual load proportion. This objective is designed to promote a more balanced load distribution across students. We evaluated this routing regularization under multiple weighting coefficients, but the empirical results (shown in the table below) indicate that incorporating $\mathcal{L}\_{\text{RR}}$ yields no meaningful performance improvement and can even lead to slight degradation. These findings demonstrate that $\mathcal{L}\_{\text{RR}}$ offers little practical benefit, which justifies our decision to omit it from Eq. 11 in the final formulation.
>
>
> | Method                              | ACM                          | Facebook                     | Reddit                       | CS                           | Photo                        | T-Finance                    | Avg.    |
> | ----------------------------------- | ---------------------------- | ---------------------------- | ---------------------------- | ---------------------------- | ---------------------------- | ---------------------------- | ------- |
> | ProMoS                              | $89.47{\scriptstyle\pm0.79}$ | $69.31{\scriptstyle\pm0.50}$ | $60.83{\scriptstyle\pm0.35}$ | $88.85{\scriptstyle\pm0.60}$ | $72.67{\scriptstyle\pm1.07}$ | $71.62{\scriptstyle\pm1.06}$ | $75.46$ |
> | + $\mathcal{L}_{\text{RR}}$ * 0.01 | $89.06{\scriptstyle\pm0.49}$ | $69.50{\scriptstyle\pm0.44}$ | $60.68{\scriptstyle\pm0.52}$ | $88.55{\scriptstyle\pm0.48}$ | $72.27{\scriptstyle\pm0.98}$ | $71.66{\scriptstyle\pm0.85}$ | $75.29$ |
> | + $\mathcal{L}_{\text{RR}}$ * 0.1  | $89.00{\scriptstyle\pm0.82}$ | $69.61{\scriptstyle\pm0.49}$ | $60.66{\scriptstyle\pm0.81}$ | $88.72{\scriptstyle\pm0.44}$ | $72.26{\scriptstyle\pm1.31}$ | $71.62{\scriptstyle\pm0.81}$ | $75.31$ |
> | + $\mathcal{L}_{\text{RR}}$ * 1    | $88.87{\scriptstyle\pm0.76}$ | $69.56{\scriptstyle\pm0.40}$ | $60.62{\scriptstyle\pm1.48}$ | $88.69{\scriptstyle\pm0.55}$ | $72.21{\scriptstyle\pm1.30}$ | $71.50{\scriptstyle\pm0.92}$ | $75.24$ |

---

> ### Author Response · Authors · 2025-11-21
> **Author Response to Reviewer 6V1N (3/3)**
>
> >W4: The authors use PubMed, Flickr, Questions and YelpChi as the pretraining datasets. Is there a specific reason? If not, what is the performance when using different pretraining datasets.
>
> **R4:** To ensure fair comparison, we adopt the ARC [3] protocol, the first to explore generalist GAD, and accordingly use PubMed, Flickr, Questions, and YelpChi for pretraining. We additionally evaluate an alternative configuration using different pretraining datasets (Alt-DPD), including Cora, CiteSeer, ACM, and PubMed. As shown in the results below, this setting achieves similarly competitive performance, demonstrating that ProMoS is **not sensitive** to the selection of pretraining datasets and **generalizes well** across different pretraining regimes. The corresponding experimental results have been added to Appendix E.5.
>
> | Method  | Facebook                     | Weibo                        | Reddit                       | CS                           | Photo                        | Tolokers                     |
> | ------- | ---------------------------- | ---------------------------- | ---------------------------- | ---------------------------- | ---------------------------- | ---------------------------- |
> | ProMoS  | $69.31{\scriptstyle\pm0.50}$ | $91.74{\scriptstyle\pm0.03}$ | $60.83{\scriptstyle\pm0.35}$ | $88.85{\scriptstyle\pm0.60}$ | $72.67{\scriptstyle\pm1.07}$ | $52.80{\scriptstyle\pm0.82}$ |
> | Alt-DPD | $68.58{\scriptstyle\pm0.38}$ | $91.77{\scriptstyle\pm0.07}$ | $58.15{\scriptstyle\pm1.58}$ | $89.15{\scriptstyle\pm0.32}$ | $73.39{\scriptstyle\pm1.15}$ | $53.96{\scriptstyle\pm0.66}$ |
>
> >W5: In Table 5, the authors argue that it reports empirical runtimes. But only the complexity of methods is provided.
>
> **R5:** We apologize for the confusion. Table 5 reports only the complexity analysis because we moved the empirical runtimes to Figures 2 and 3, where bar plots allow comparison against a larger set of baselines. For clarity, we also provide the corresponding training and inference runtimes for zero-shot generalist GAD methods in the table below, showing that ProMoS remains competitively efficient in both stages.
>
> | Method     | Train (s) | Infer (s) |
> | ---------- | --------- | --------- |
> | ProMoS     | 1.87      | 10.25     |
> | UNPrompt   | 268.32    | 98.63     |
> | AnomalyGFM | 8329.67   | 135.82    |
>
> [1] Hou, Zhenyu, et al. "Graphmae: Self-supervised masked graph autoencoders." KDD 2022.
> [2] Liu, Yixin, et al. "Graph self-supervised learning: A survey." TKDE 2022.
> [3] Liu, Yixin, et al. "Arc: A generalist graph anomaly detector with in-context learning." NeurIPS 2024.

---

> ### Author Response · Authors · 2025-11-27
>
> Dear Reviewer 6V1N,
>
> We would like to express our sincere gratitude to you for reviewing our paper and providing valuable feedback. We believe that we have responded to and addressed all your concerns with our revisions — in light of this, we hope you consider raising your score.
>
> If you have any additional questions or would like further clarification on any point, we would be very glad to respond promptly. Thanks!
>
> Best,
>
> All authors

---

### Official Review · Reviewer_6gsy · 2025-10-30

**Soundness:** 3
**Presentation:** 3
**Contribution:** 3
**Rating:** 6
**Confidence:** 4

**Summary:**

This paper presents an unsupervised generalist GAD framework, ProMos, which transfers prior knowledge from a pre-trained graph self-supervised learning teacher and introduces MOS to balance expressiveness and efficiency. The framework is jointly optimized using a tailored set of loss functions, including prototype distillation, as well as discrepancy-aware commitment and refinement losses.

**Strengths:**

(1) This paper is well-motivated and well-written. Distilling knowledge from a pre-trained SSL model is an effective approach that aligns well with intuitive understanding.

(2) The authors propose a fine-grained, prototype-guided method that goes beyond the conventional binary classification of normal and abnormal classes.

**Weaknesses:**

(1) The details regarding the pre-training of the teacher model are not very detailed, particularly how the training across multiple graphs is integrated with the clustering process. There are four graph inputs—do the authors directly merge all nodes from these graphs and then perform clustering on the combined set?

(2) It is also unclear whether the inputs to the student model are identical to those of the teacher model. The distillation process is intended to learn invariant features across different graphs, where the guidance from prototype learning helps the model capture the underlying normal patterns.

(3) Could the authors include a t-SNE visualization for one of the datasets to illustrate the effectiveness of the commitment loss and refinement loss? It would also be helpful to show the difference between the teacher’s and the student’s feature distributions.

**Questions:**

See above **Weaknesses**

---

> ### Author Response · Authors · 2025-11-21
> **Author Response to Reviewer 6gsy**
>
> We appreciate your positive feedback on the motivation, writing clarity, and effectiveness of our method design. Next we will reply to comments one by one.
>
> > W1: The details regarding the pre-training of the teacher model are not very detailed, particularly how the training across multiple graphs is integrated with the clustering process. There are four graph inputs—do the authors directly merge all nodes from these graphs and then perform clustering on the combined set?
>
> **R1:** We apologize for the confusion arising from the insufficient description. For teacher pre-training across multiple graphs, we follow established protocols [1, 2]: the SSL GNN is trained *sequentially* on the four training graphs within each epoch, rather than merging them into a single large graph. For prototype initialization, clustering is indeed performed on the concatenated node features from the four training graphs using FAISS k-means. Additional implementation details have been added to Appendix D.2 for completeness.
>
> >W2: It is also unclear whether the inputs to the student model are identical to those of the teacher model. The distillation process is intended to learn invariant features across different graphs, where the guidance from prototype learning helps the model capture the underlying normal patterns.
>
> **R2:** Thank you for pointing this out, we would like to note that the teacher and student models **do not share** the same inputs. As illustrated in Figure 1, the teacher receives the full graph input, comprising the adjacency matrix and node feature matrix, as defined in Eq. (1). In contrast, the student operates *only* on node features augmented with residual information, as defined in Eq. (2). Prototype-guided distillation enables the student to “stand on the shoulders” of the teacher, efficiently acquiring invariant patterns that generalize well to unseen graphs.
>
> >W3: Could the authors include a t-SNE visualization for one of the datasets to illustrate the effectiveness of the commitment loss and refinement loss? It would also be helpful to show the difference between the teacher’s and the student’s feature distributions.
>
> **R3:** In the revised manuscript, we have added t-SNE visualizations in Appendix E.6. Specifically:
>
> - **Effectiveness of the commitment and refinement losses.** Figures 5(a) and 5(b) compare the teacher outputs on the unseen Cora graph. Without $\mathcal{L}\_{\text{DCR}}$, the teacher features become scattered, and the prototypes lose coherent semantic structure under distribution shift. When $\mathcal{L}\_{\text{DCR}}$ is applied, both the shared and personalized prototypes remain representative and more discriminative, and anomalous nodes are pushed away from prototype-dense regions. These observations confirm that DCR effectively optimizes the prototype space and regularizes the teacher representation space, thereby improving the robustness of teacher features on unseen graphs.
> - **Teacher–student feature distributions.** Figure 5\(c) visualizes the teacher and student embeddings on CiteSeer and shows that their overall distributions remain well aligned, indicating that the student network can approximate the teacher’s representation space even under distribution shift.
>
> We further conduct a fine-grained analysis by randomly sampling 20 normal and 20 anomalous teacher–student pairs in  Figure 5\(c), where each pair consists of a node’s teacher embedding and its corresponding student embedding. We then visualize the line segment connecting the two embeddings in the projected space. **Normal pairs show short distances and strong alignment**, indicating that MoS reliably reconstructs teacher representations. Anomalous pairs show much larger deviations due to irregular semantics and weak prototype affinity, leading to poor reconstruction—consistent with our distillation-bias anomaly score. Additional details and visualizations are provided in Appendix E.6.
>
> [1] Liu, Yixin, et al. "Arc: A generalist graph anomaly detector with in-context learning." NeurIPS 2024.
>
> [2] Dong, Kaiwen, et al. "Universal link predictor by in-context learning on graphs." arXiv preprint.

---

> ### Author Response · Authors · 2025-11-27
>
> Dear Reviewer 6gsy,
>
> We would like to express our sincere gratitude to you for reviewing our paper and providing valuable feedback. We believe that we have responded to and addressed all your concerns with our revisions — in light of this, we hope you consider raising your score.
>
> If you have any additional questions or would like further clarification on any point, we would be very glad to respond promptly. Thanks!
>
> Best,
>
> All authors

---

> > ### Comment · Reviewer_6gsy · 2025-11-27
> >
> > Thanks for the detailed rebuttal. The response has addressed most of my concerns, and I will maintain my **positive score** for this paper.

---

### Official Review · Reviewer_RTkz · 2025-10-31

**Soundness:** 2
**Presentation:** 3
**Contribution:** 2
**Rating:** 4
**Confidence:** 4

**Summary:**

This paper presents ProMoS, which aims to perform graph anomaly detection in a general and unsupervised manner. The approach uses a frozen self-supervised GNN as the teacher to provide representations, while a mixture-of-students model learns to capture multiple normality patterns through prototype-based soft supervision and discrepancy-aware refinement. Experiments on several benchmark graphs show that the method achieves strong zero-shot detection performance and demonstrates good generalization across different graph domains.

**Strengths:**

1. The motivation is novel and meaningful, as it considers the diverse anomaly patterns that exist across different graph datasets.
2. The overall framework and training objectives are clear and well explained.
3. The figures and layout are well organized and easy to follow, and the writing is clear with comprehensive experiments.

**Weaknesses:**

1. The method does not effectively ensure diversity among student models. Although multiple students are used to capture different modes, no concrete strategy enforces their differentiation.
2. Some formula notations are incorrect, such as in Eq. (2), where ‘ei’ and ‘ri’ appear inconsistent and likely refer to the same variable.
3. The ablation study shows limited improvement from modules PB, SB, and DIS, indicating their contributions are not significant.
4. The Discrepancy-aware Commitment and Refinement stage is somewhat confusing. It is unclear whether simultaneously adjusting the teacher embeddings and prototype vectors might weaken the distinctiveness and effectiveness of the prototypes.

**Questions:**

Please see the weaknesses.

---

> ### Author Response · Authors · 2025-11-21
> **Author Response to Reviewer RTkz (1/3)**
>
> We sincerely thank you for your constructive feedback and positive remarks on our novel motivation, framework clarity, and comprehensive experimental. Next we will reply to comments one by one.
>
> >W1: The method does not effectively ensure diversity among student models. Although multiple students are used to capture different modes, no concrete strategy enforces their differentiation.
>
> **R1:** The diversity among student models in ProMoS is structurally **encouraged through the sparse Top-K routing mechanism**. Top-K routing assigns each node to only a subset of students based on its representation. This routing structure naturally steers different students toward different subsets of data, fostering specialization without requiring an explicit diversity regularizer.
>
> We further computes the activation frequency of each student during inference, showing that **different inputs tend to activate different students**, rather than relying on a single dominant student. We also provide visualizations in Appendix E.6 based on student-specific embeddings from the Weibo dataset (Figure 5). The resulting clusters are clearly separated across students, indicating that each student focuses on distinct semantic patterns of the feature space. Overall, these observations demonstrate that ProMoS achieves meaningful diversity among student models in practice.
>
> | Students | 1      | 2      | 3      | 4      | 5      | 6      | 7      | 8      | 9      | 10     | 11     | 12     | 13     | 14     | 15     | 16     | 17     | 18     | 19     | 20     |
> | -------- | ------ | ------ | ------ | ------ | ------ | ------ | ------ | ------ | ------ | ------ | ------ | ------ | ------ | ------ | ------ | ------ | ------ | ------ | ------ | ------ |
> | Cora     | 0.0290 | 0.0486 | 0.1418 | 0.0319 | 0.0013 | 0.0545 | 0.1165 | 0.0299 | 0.0175 | 0.0024 | 0.0089 | 0.0971 | 0.0199 | 0.0663 | 0.0801 | 0.1451 | 0.0735 | 0.0210 | 0.0002 | 0.0144 |
> | CiteSeer | 0.0397 | 0.0353 | 0.1551 | 0.0431 | 0.0018 | 0.0397 | 0.1220 | 0.0370 | 0.0334 | 0.0021 | 0.0114 | 0.0818 | 0.0274 | 0.0562 | 0.0810 | 0.1392 | 0.0604 | 0.0108 | 0.0002 | 0.0225 |
> | Facebook | 0.0541 | 0.0171 | 0.0005 | 0.0365 | 0.0694 | 0.0009 | 0.0079 | 0.0560 | 0.0069 | 0.1443 | 0.0620 | 0.0301 | 0.0069 | 0.0023 | 0.1369 | 0.1013 | 0.0000 | 0.0005 | 0.0865 | 0.1799 |
>
>
> Although the students already exhibit a degree of diversity, we acknowledge that an explicit strategy to further encourage it could be beneficial and evaluat a simple diversity regularization term:
> $
> \mathcal{L}\_{\text{DIV}}= -\frac{1}{2|\mathcal{V}|}\sum\_{i\in\mathcal{V}}\Big[\mathrm{KL}\\!\left(\mathbf{h}\_{i,1}^{\ell}\,\big\Vert\,\mathbf{h}\_{i,2}^{\ell}\right)+\mathrm{KL}\\!\left(\mathbf{h}\_{i,2}^{\ell}\,\big\Vert\,\mathbf{h}\_{i,1}^{\ell}\right)\Big]
> $
> , where $\mathbf{h}\_{i,1}^{\ell}$ and $\mathbf{h}\_{i,2}^{\ell}$ denote the outputs of the two most highly activated students routed for node $i$ according to Eq. (4).
>
> As shown in the table below, the regularizer leads to improvements on some datasets (e.g., T-Finance), but does not yield consistent gains across all settings. A possible explanation is that the sparse Top-K routing already provides sufficient functional diversity, additional separation constraints may offer limited benefit or even interfere with the teacher–student prototype distillation objective. It suggests that explicit diversity promotion is promising in principle but that a naive instantiation is insufficient. Therefore, we regard the careful design of more principled, task-aware diversity mechanisms as a promising direction for future work.
>
> | Method                              | ACM                          | Facebook                     | Reddit                       | CS                           | Photo                         | T-Finance                    | Avg.             |
> | ----------------------------------- | ---------------------------- | ---------------------------- | ---------------------------- | ---------------------------- | ----------------------------- | ---------------------------- | ---------------- |
> | ProMoS                              | $89.47{\scriptstyle\pm0.79}$ | $69.31{\scriptstyle\pm0.50}$ | $60.83{\scriptstyle\pm0.35}$ | $88.85{\scriptstyle\pm0.60}$ | $72.67{\scriptstyle\pm1.07}$  | $71.62{\scriptstyle\pm1.06}$ | $\mathbf{75.46}$ |
> | + $\mathcal{L}_{\text{DIV}}$ * 0.01 | $88.94{\scriptstyle\pm1.22}$ | $69.32{\scriptstyle\pm0.46}$ | $60.81{\scriptstyle\pm0.29}$ | $88.70{\scriptstyle\pm0.74}$ | $72.66{\scriptstyle\pm0.97}$  | $71.79{\scriptstyle\pm1.03}$ | 75.37            |
> | + $\mathcal{L}_{\text{DIV}}$ * 1    | $63.60{\scriptstyle\pm5.75}$ | $55.17{\scriptstyle\pm3.73}$ | $53.11{\scriptstyle\pm2.72}$ | $61.47{\scriptstyle\pm9.43}$ | $46.26{\scriptstyle\pm10.32}$ | $72.17{\scriptstyle\pm4.51}$ | 58.63            |

---

> ### Author Response · Authors · 2025-11-21
> **Author Response to Reviewer RTkz (2/3)**
>
> >W2: Some formula notations are incorrect, such as in Eq. (2), where ‘ei’ and ‘ri’ appear inconsistent and likely refer to the same variable.
>
> **R2:** Thank you for highlighting this issue. In Eq. (2), $\mathbf{e}_i$ was mistakenly written as $\mathbf{r}_i$. The correct residual update is $\tilde{\mathbf{x}}_i = \mathbf{x}_i + \mathbf{e}_i$. The symbol $\mathbf{r}_i$ is used only later in Eq. (4) to denote the routing probability of node $i$. We have corrected this notation in the revised manuscript and carefully reviewed the entire method section to prevent similar inconsistencies.
>
> >W3: The ablation study shows limited improvement from modules PB, SB, and DIS, indicating their contributions are not significant.
>
>
> **R3:** Although removing PB, SB, or DIS individually leads to only modest changes, **these components do not operate independently**. SB and PB together form the Mixture of Students (MoS), capturing complementary global and local normality patterns. DIS assigns reliability-aware weights to the commitment and refinement objectives, steering prototype updates toward high-quality semantic structure. These **modules act synergistically**, and their benefits are clearer under joint evaluation.
>
> To make this interaction explicit, we added **joint ablations** that remove DIS together with PB or with SB. As shown in the table below, removing PB + DIS or SB + DIS leads to substantial drops on all four datasets. On ACM, the decreases are 9.80% and 9.20%, and on CS and Tolokers both variants reduce performance by more than 6%. These declines are far larger than those observed in the single-module ablations, confirming that PB, SB, and DIS offer complementary functionality and strengthen one another when used together.
>
> Taken together, the joint and single ablations show that PB, SB, and DIS address different facets of the generalist GAD challenge and that **their interaction is critical for the robustness and stability of ProMoS**.
>
> | Method           | ACM                                                          | Reddit                                                       | CS                                                           | Tolokers                                                     |
> | ---------------- | ------------------------------------------------------------ | ------------------------------------------------------------ | ------------------------------------------------------------ | ------------------------------------------------------------ |
> | ProMoS           | $89.47{\scriptstyle\pm0.79}$                                 | $60.83{\scriptstyle\pm0.35}$                                 | $88.85{\scriptstyle\pm0.60}$                                 | $52.80{\scriptstyle\pm0.82}$                                 |
> | w/o PB + w/o DIS | $79.67{\scriptstyle\pm0.64}\left ( \downarrow_{9.80} \right )$ | $55.98{\scriptstyle\pm2.11}\left ( \downarrow_{4.85} \right )$ | $82.02{\scriptstyle\pm0.26}\left ( \downarrow_{6.83} \right )$ | $45.93{\scriptstyle\pm0.85}\left ( \downarrow_{6.87} \right )$ |
> | w/o SB + w/o DIS | $80.27{\scriptstyle\pm0.44}\left ( \downarrow_{9.20} \right )$ | $56.31{\scriptstyle\pm0.86}\left ( \downarrow_{4.52} \right )$ | $82.18{\scriptstyle\pm0.75}\left ( \downarrow_{6.67} \right )$ | $45.57{\scriptstyle\pm1.07}\left ( \downarrow_{7.23} \right )$ |

---

> ### Author Response · Authors · 2025-11-21
> **Author Response to Reviewer RTkz (3/3)**
>
> > W4: The Discrepancy-aware Commitment and Refinement stage is somewhat confusing. It is unclear whether simultaneously adjusting the teacher embeddings and prototype vectors might weaken the distinctiveness and effectiveness of the prototypes.
>
> **R4:** Thank you for pointing this out. We have substantially revised and clarified the discrepancy-aware commitment and refinement component. Specifically, we:
>
> - **Clarify the objective’s role**: The commitment and refinement objective (Eq. 10) regularizes the teacher outputs to enforce a consistent and well-structured feature space across graphs, and simultaneously updates the prototypes to capture high-level, transferable semantics.
>
> - **Explain the reliability-aware weighting strategy**: Eq. 9 implements an adaptive weighting mechanism that assigns larger weights to high-quality samples and downweights unreliable ones. In the revised manuscript, we provide additional details and a clearer explanation of the intuition behind this design.
>
> - **Detail the loss design**: We define discrepancy-aware commitment and refinement losses, where the commitment loss pulls teacher features toward prototypes and the refinement loss updates prototypes to better reflect transferable semantic structure.
>
> We provide a clearer, step-by-step explanation of this mechanism in the revised subsection.
>
> To assess the influence *simultaneously* updating the teacher embeddings and prototypes, we further evaluated two *sequentially* updating variants: (1) updating the prototypes then the teacher embeddings (Alt-PT), and (2) updating the teacher embeddings then the prototypes (Alt-TP). All three strategies lead to consistent performance, and simultaneous optimization yields slightly better results, indicating that jointly updating teachers and prototypes **does not weaken prototype distinctiveness** and remains a robust design choice. We further provide visualization in Appendix E.6 indicating that the prototypes' semantic distinctiveness is effectively preserved.
>
>
> | Method         | ACM                          | Facebook                     | Reddit                       | CS                           | Photo                        | T-Finance                    | Avg.             |
> | -------------- | ---------------------------- | ---------------------------- | ---------------------------- | ---------------------------- | ---------------------------- | ---------------------------- | ---------------- |
> | Alt-PT         | $89.67{\scriptstyle\pm1.00}$ | $69.19{\scriptstyle\pm0.57}$ | $60.73{\scriptstyle\pm0.66}$ | $88.95{\scriptstyle\pm1.08}$ | $72.80{\scriptstyle\pm1.36}$ | $70.58{\scriptstyle\pm0.65}$ | 75.32            |
> | Alt-TP         | $89.86{\scriptstyle\pm0.65}$ | $68.95{\scriptstyle\pm0.50}$ | $60.62{\scriptstyle\pm0.98}$ | $89.16{\scriptstyle\pm0.82}$ | $71.19{\scriptstyle\pm1.69}$ | $70.39{\scriptstyle\pm1.15}$ | 75.03            |
> | ProMoS (Joint) | $89.47{\scriptstyle\pm0.79}$ | $69.31{\scriptstyle\pm0.50}$ | $60.83{\scriptstyle\pm0.35}$ | $88.85{\scriptstyle\pm0.60}$ | $72.67{\scriptstyle\pm1.07}$ | $71.62{\scriptstyle\pm1.06}$ | $\mathbf{75.46}$ |

---

> ### Author Response · Authors · 2025-11-27
>
> Dear Reviewer RTkz,
>
> We would like to express our sincere gratitude to you for reviewing our paper and providing valuable feedback. We believe that we have responded to and addressed all your concerns with our revisions — in light of this, we hope you consider raising your score.
>
> If you have any additional questions or would like further clarification on any point, we would be very glad to respond promptly. Thanks!
>
> Best,
>
> All authors

---

### Author Response · Authors · 2025-11-29
**General Response to All Reviewers**

We sincerely thank all the reviewers and chairs for their efforts. During the rebuttal and discussion phase, we have received strong support and positive feedback on our paper:


- The studied problem is **challenging and practically important** (`Reviewer 6V1N, tmaC`)

- The proposed method is **novel, effective, and meaningful** (`Reviewer RTkz, 6gsy, 6V1N, tmaC`)

- The paper is **well-written with clear logic and structure** (`Reviewer RTkz, 6gsy, tmaC`)

- The experiments are **comprehensive** and demonstrate **strong effectiveness** (`Reviewer RTkz, 6V1N, tmaC`)

- The released code enhances **reproducibility and transparency** (`Reviewer 6V1N, tmaC`)


Beyond these positive feedbacks, the reviewers provided many constructive suggestions that significantly helped improve the quality of the paper. Accordingly, we have made several substantial improvements:

- **Significantly revised and clarified** the discrepancy-aware commitment and refinement module (Lines 272–309).

- **Corrected** the notation error in Eq. 2 (Line 213).

- **Added detailed descriptions** of the model’s pretraining procedure (Lines 1093–1098).

- Introduced a **reconstruction-based variant** as an alternative teacher model (Appendix E.3, Lines 1176–1187).

- Compared **alternative update strategies** for the commitment and refinement objectives (Appendix E.4, Lines 1206–1230).

- Reported performance across **different pretraining datasets** (Appendix E.5, Lines 1232–1247).

- Incorporated four complementary **visualization analyses** to enhance interpretability (Appendix E.6, Lines 1250–1315).

We believe these enhancements address the reviewers’ concerns and substantially reinforce the clarity, robustness, and completeness of the paper.

---

### Meta-Review · Area_Chair_DbEB · 2026-01-06

**Summary:**

This paper proposes ProMoS, an unsupervised generalist graph anomaly detection framework that performs zero-shot detection on unseen graphs via prototype-guided teacher–student distillation with a mixture-of-students architecture.

Reviewers agree that the problem is important and timely, and that the overall framework is well motivated, clearly written, and empirically strong, with comprehensive experiments and released code. However, several core weaknesses remain. Multiple reviewers raised concerns about methodological clarity and correctness, including notation errors, unclear teacher pretraining and clustering across graphs, ambiguous teacher–student input differences, and confusion around the discrepancy-aware commitment and refinement mechanism. While the rebuttal provides clarifications and additional appendix analyses, these explanations remain complex and do not fully resolve concerns about the necessity, stability, and interpretability of the key components. In particular, student diversity in the MoS is not explicitly enforced, and ablation results suggest that individual modules provide limited standalone gains, weakening the strength of the claimed contributions.

Overall, despite promising performance, the paper does not yet provide sufficiently clear, principled, and convincing evidence for its central design choices. Therefore, I recommend rejection.

**Reviewer Concerns:**

-  The principled enforcement of diversity in the mixture-of-students.
-  The limited standalone impact of individual modules as shown by ablations.
-  The overall complexity and interpretability of the commitment/refinement design.

**Reviewer Scores:**

Reviewer RTkz would like increase the score slightly,  since several technical issues and ablation concerns were addressed.

---

### Decision · Program_Chairs · 2026-01-26

Reject